# Secure Monitoring System for IoT Healthcare Data in the Cloud

Christos L. Stergiou *, Andreas P. Plageras, Vasileios A. Memos, Maria P. Koidou
and Konstantinos E. Psannis *

Department of Applied Informatics, University of Macedonia, 54636 Thessaloniki, Greece;
a.plageras@uom.edu.gr (A.P.P.); vmemos@uom.edu.gr (V.A.M.); maria_koidou@yahoo.gr (M.P.K.)
* Correspondence: c.stergiou@uom.edu.gr (C.L.S.); kpsannis@uom.edu.gr (K.E.P.); Tel.: +30-2310-891-737 (K.E.P.)

**Abstract:** Even though the field of medicine has made great strides in recent years, infectious diseases caused by novel viruses that damage the respiratory system continue to plague people all over the world. This type of virus is very dangerous, especially for people with serious long-term breathing problems like asthma, pneumonia, or bronchitis infections. Thus, this paper demonstrates a new secure machine learning monitoring system for a model for virus detection. Our proposed model makes use of four basic emerging technologies, the Internet of Things (IoT), Wireless Sensor Networks (WSN), Cloud Computing (CC), and Machine Learning (ML), to detect dangerous types of viruses that infect people or animals causing panic worldwide and deregulating human daily life. The proposed system is a robust system that could be established in various buildings, like hospitals, entertainment halls, universities, etc., and will provide accuracy, speed, and privacy for data collected in the detection of viruses.

**Keywords:** internet of things; machine learning; cloud computing; artificial intelligence; security; healthcare; monitoring; detection





## 1. Introduction

It appears that there is a pressing need to incorporate new technologies into medical care for the best possible detection of such viruses. Five cutting-edge technologies—Internet of Things (IoT), Wireless Sensor Networks (WSN), Cloud Computing (CC), Machine Learning (ML), and Wireless Networks—can be utilized in conjunction to combat viruses and the threat of infectious illnesses. All these technologies can be worked under the broad term of Internet of Medical Things (IoMT), which is the IoT that has reconstructed hospital settings and invented a new paradigm. Additionally, it offers numerous opportunities, due to the wearable devices used by plenty of people, to enhance health and well-being associated with eHealth and mHealth [1]. WSN takes advantage of features such as low cost, availability, and accessibility, and as a result, there has been an increase in the adoption of these mobile sensors. Additionally, these personalized healthcare systems gather pertinent biophysical data to aid in medical diagnoses and decisions, and they could be stored, maintained, and processed in the cloud [2].

Moreover, IoT-based Big Data (BD) and ML can offer a handful of opportunities for healthcare systems relying on IoT, and therefore the new term of Artificial Intelligence of Things (AIoT). WSN systems' real-time health data can be utilized to assist patients with self-administered treatments. Furthermore, mobile devices with mobile applications are frequently used and integrated with the terms of telemedicine and mHealth through the IoMT. The results of medical data analytics from data analysis platforms established on the cloud increase the applicability of data interpretations and reduce the time of analyzing data outputs and, thus, the detection and prediction of viruses and diseases.

A secure monitoring system for IoT healthcare data in the cloud needs to be designed to ensure the confidentiality, integrity, and availability of sensitive healthcare data generated by Internet of Things (IoT) devices [1,2]. It must aim to protect patient privacy, prevent

unauthorized access, and maintain the integrity of healthcare data stored and processed in the cloud. Below is an overview of the implementation aspects and goals of such a system: (1) *Data encryption*: Implement strong encryption techniques to secure healthcare data both in transit and at rest. Encryption ensures that even if the data is intercepted or accessed without authorization, it remains unreadable and useless. (2) *Access control and authentication*: Enforce strict access control mechanisms to allow only authorized personnel or systems to access healthcare data. This includes implementing strong authentication methods like two-factor authentication and role-based access control (RBAC) to limit data access based on user roles and responsibilities. (3) *Secure communication*: Establish secure channels for communication between IoT devices, cloud infrastructure, and healthcare applications. This involves using secure protocols such as Transport Layer Security (TLS) or Secure Sockets Layer (SSL) to encrypt data in transit and prevent eavesdropping or tampering. (4) *Data integrity and validation*: Implement mechanisms to ensure the integrity of healthcare data. This can be achieved through digital signatures or message authentication codes (MACs), which verify that the data has not been tampered with during transit or storage. (5) *Security monitoring and logging*: Set up monitoring systems to detect any suspicious activities or unauthorized access attempts. Log and analyze system events, network traffic, and user activities to identify potential security breaches or anomalies. (6) *Secure storage*: Utilize secure storage solutions to protect healthcare data within the cloud infrastructure. This may involve using encryption at rest, access controls, and redundancy to ensure data availability and resilience against hardware failures or cyber-attacks. (7) *Compliance with regulations*: Ensure compliance with relevant healthcare regulations such as the Health Insurance Portability and Accountability Act (HIPAA) or the General Data Protection Regulation (GDPR). Adhere to the necessary security and privacy requirements specified by these regulations. (8) *Regular security audits and updates*: Perform regular security audits and assessments to identify vulnerabilities and implement necessary updates and patches. Stay up to date with the latest security practices, emerging threats, and industry standards.

Therefore, the main contributions of this work are:

- The proposed system integrates the benefits of IoT with the computational power of cloud computing through a wireless network.
- The proposed model immediately detects, predicts, and notifies the responsible surveillance personnel.
- The proposed model could be established in public service buildings, such as hospitals.
- The proposed system has a high transmission data rate based on its operation over a wireless network and additionally aims to have more direct notifications due to the IoT data produced by mobile devices.

The rest of the paper is organized as follows: Section 2 presents some related research conducted in the field of virus detection. Section 3 analyzes the limitations and barriers of conventional medical methods. Section 4 describes the technological challenges generated in microbiology. Section 5 describes our proposed model and the involved technologies. Section 6 presents the application fields of our proposed scheme and the benefits that it can provide to the population worldwide. Finally, Section 7 concludes the paper and provides some potential future directions.

## 2. Related Work

The field of medicine has offered numerous studies that use various technologies to detect harmful viruses and contagious diseases effectively. The development of an autonomous robust model, though, is still being investigated. This model will utilize all of the most well-known immersive technologies.

The objective of the research by L. Bai et al. [3] was to timely diagnose patients and provide the best possible care using medical technology, namely the "COVID-19 Intelligent Diagnosis and Treatment Assistant Program (nCapp)". Real-time communication is facilitated by computer cloud technology, which is a feature of nCapp. This study shows that when a suspicious sample is found, the diagnosis is automated, and nCapp then categorizes

the suspects based on how serious the problem is. The system is updated in real-time by nCapp, which also makes it a reliable resource for illnesses in the future.

H. S. Maghdid [4] has developed a framework for COVID-19 detection using information from a smartphone's onboard sensors, including cameras, microphones, temperature, and inertial sensors. Based on the gathered data, ML techniques are used to learn about and gain information about the disease symptoms. This is perhaps understandable given that the data generated by the smartphone sensors has already been successfully used in a number of distinct applications, and the suggested solution combines all of these applications into a single framework.

S. Muthukumar [5] suggests a smart humidity monitoring system that takes into account the strong connection between humidity and infectious disease and suggests a method that may maintain a room's relative humidity while preventing the spread of infectious diseases. A sensor-based IoT module has been developed that monitors a room's relative humidity and provides the inhabitants with its status after considering the benefits of preserving humidity. Specifically, the sensor-based hardware module measures the air's relative humidity and provides the data to the user through the internet, whether they are in the same room as the user or not. The aforementioned module can also be set to maintain relative humidity levels in a room depending on the user's requirements. In order to maintain ideal humidity levels, the system also manages the room's air conditioning and humidifying equipment. Due to the significant danger of infectious disease outbreaks in hospitals, the designed module is an affordable option that can be very helpful. It can be used in homes, offices, and schools due to its inexpensive cost.

In their study, V. K. Quy et al. [6] try to identify a complete picture of changes in architectures, technologies, and challenges that will shape the 6G network. Their experimental results provide indications for further studies on 6G ecosystems.

Furthermore, in a different study, V. K. Quy et al. [7] suggested an all-in-one computer architecture framework. This framework for the Internet of Health Things (Fog-IoHT) apps is fog computing-based. Based on the findings of this study, the authors also suggest potential uses and difficulties in incorporating fog computing into IoT healthcare applications.

## 3. Limitations and Barriers of Conventional Medical Methods

Although conventional medical methods and techniques play an important role in fighting against viruses and infectious diseases caused by them, there are several limitations and barriers related to the identification and, hence, the treatment of fatal diseases, which are very dangerous for the human population.

The branch of medicine, especially microbiology, is evolving continuously, using novel techniques for enhanced healthcare results. However, the integration of technological methods in microbiology is still in the early stages. The human population still suffers from fatal diseases caused by dangerous viruses that insult the respiratory system. Such viruses, like the new Coronavirus COVID-19, are very hazardous for everyone, especially those who face serious long-term breathing problems like asthma, pneumonia, or bronchitis infections.

The medical sector has to deal with infectious diseases and several limitations and barriers, such as the lack of health personnel and health infrastructure (medical centers, hospitals, etc.), in combination with the difficult organizational healthcare of emergency cases [8], like nowadays with the COVID-19 spread. These problems make the fight against sudden epidemics, such as those of recent days, very difficult, causing many delays in virus identification and treatment of infectious diseases and, in many cases, deaths. In addition, this weakness of the medical sector also impacts the economy, as most businesses have been forced to close, and thus, many employees were left without work and salary.

*Limitations Could Be Addressed by IoT*

Conventional medical methods have been effective in treating various health conditions, but they do have some limitations and barriers that can be addressed by integrating

the Internet of Things (IoT) and related technologies [9–11]. Some of these limitations and how the IoT can help overcome them are [12,13]:

*Limited monitoring:* Conventional medical methods often involve periodic visits to healthcare facilities, which can lead to limited monitoring of patients' health conditions. IoT-enabled devices, such as wearable health trackers and remote patient monitoring systems, offer continuous and real-time health data collection, providing healthcare professionals with a more comprehensive view of patients' health statuses.

*Inefficient data collection:* The manual recording of patient information used in traditional data-gathering techniques in the healthcare industry can be time-consuming and prone to inaccuracy. By automating data gathering procedures, IoT technologies lower the possibility of human error and facilitate quicker access to vital health data.

*Lack of personalization:* Conventional medical practices frequently rely on generic methods that may not adequately address the needs of each patient. By evaluating a patient's real-time data, the IoT can enable personalized medicine by allowing healthcare professionals to customize interventions and treatments depending on particular health problems and response patterns.

*Limited access to healthcare in remote areas:* There are many localities with poor access to healthcare facilities, particularly in rural or outlying areas. By enabling virtual consultations, remote diagnostics, and ongoing monitoring, IoT-enabled telemedicine and remote healthcare systems help close this gap, enhancing healthcare access and results in underserved areas.

*Delayed diagnosis and intervention:* Particularly in the case of chronic illnesses, traditional healthcare approaches may result in delayed diagnosis and action. IoT-based remote monitoring enables early abnormality detection, prompt response, and the avoidance of potential health emergencies.

*Fragmented healthcare systems:* Traditional medical practices frequently use disjointed data storage and communication platforms, which makes it difficult to access thorough patient records. Using IoT technologies, a more interconnected and unified healthcare system may be created by combining data from many sources and devices.

*Patient compliance and adherence:* It might be difficult to monitor and enforce patient adherence to recommended therapies and pharmaceutical regimens. IoT gadgets can deliver alerts, monitor medicine intake, and give behavioral cues, encouraging patients to follow their treatment regimens.

*Limited data for research and analysis:* For medical research and analysis, traditional medical practices might only supply partial data. Big data analytics, medical research, and the creation of prognostic models for illness prevention and treatment are all greatly facilitated by IoT-generated data.

*Data security and privacy concerns:* IoT use in healthcare sparks worries about patient privacy and data security. Strong data encryption, secure communication methods, and adherence to pertinent data protection laws like HIPAA and GDPR are all necessary for addressing these issues.

Healthcare may overcome these restrictions and obstacles by utilizing the IoT and related technologies, which will result in better patient outcomes, improved disease management, and more effective healthcare delivery. When implementing IoT solutions in the healthcare industry, it is crucial to solve issues with data protection, interoperability, and scalability.

## 4. Technological Challenges in Microbiology

Previous works [14–17] have demonstrated that the rapid development of wired and mobile networks has resulted in a rapid growth in data. Therefore, managing, analyzing, and transferring these kinds of data presents us with significant hurdles. These challenges regarding the vast amount of data are related to their representation, expendability, existing redundancy, quality and variety, storage, knowledge exported from them, the management of their life cycle, energy management, heterogeneity, speed and accuracy, security and

privacy, confidentiality, the generation or development of tools, methods, and algorithms for analysis, and the overall performance of them [18]. The most recent research shows that there are "gaps" in the way that such data are managed, analyzed, and transported at various levels, as well as issues that arise from their use. As a result, it is necessary to optimize them by applying new techniques and algorithms [16]. Moreover, state mechanisms of many countries, mainly the ones of third countries, are not ready to apply cutting-edge technologies to their population.

*Technological Challenges in Microbiology Addressed by IoT*

IoT (Internet of Things) and associated technologies can be very helpful in addressing several technological issues in the microbiology sector. The study of microorganisms, such as bacteria, viruses, fungi, and other tiny organisms, is known as microbiology [9,12,19]. The IoT can assist in overcoming the following technological issues in microbiology [11,13]:

*Real-time monitoring of microbial growth:* Traditional techniques for keeping track of microbial development in cultures can be labor-intensive and time-consuming. Researchers can remotely monitor and control environmental conditions thanks to the ability of IoT-enabled sensors to offer real-time data on variables like temperature, pH, and oxygen levels.

*Remote data collection from field samples:* It can be logistically difficult to gather field samples for microbiological investigation and transfer them to the lab. Researchers may track microbial activity on-site using IoT-based remote sensing devices to collect data from varied environments and communicate it instantaneously to labs.

*Data-intensive microbiome analysis:* Microbiome analysis, which examines microbial communities and their interactions throughout various ecosystems, produces significant data. High-throughput sequencing technology and IoT-based sensors can speed up data gathering and processing, which will improve our understanding of complex microbiomes.

*Early detection of infectious diseases:* For the sake of public health, rapid and early detection of infectious diseases is essential. IoT devices can identify specific microbiological indicators, facilitating quicker diagnosis and intervention. Examples include wearable biosensors and point-of-care diagnostic tools.

*Environmental monitoring for outbreaks:* It might be difficult to monitor environmental conditions that cause illness epidemics, like air pollution and water quality. In order to identify potential outbreak hazards, the IoT sensors may continuously monitor these elements and give real-time data.

*Laboratory automation and efficiency:* Microbiology laboratory processes can be labor and time-intensive. Lab automation and IoT-connected hardware can automate sample processing, data collection, and analysis, increasing accuracy and efficiency.

*Remote collaboration and data sharing:* The requirement for physical presence in the lab can make it difficult for researchers to work together on microbiology projects in various places. The IoT technologies make it possible for researchers to access and exchange data online, facilitating remote collaboration securely.

*Patient monitoring and management*: In medical microbiology, it is important to keep an eye on individuals who have infections. IoT devices can help with ongoing patient monitoring by sending information to healthcare professionals regarding infection signs, the efficacy of treatments, and the progression of recovery.

*Antibiotic resistance surveillance:* Antibiotic resistance is being tracked and addressed, which is a challenge for world health. Real-time monitoring of antimicrobial use and resistance patterns by IoT-enabled equipment can help with surveillance and well-informed decision-making.

*Quality control in the food and beverage industry*: Testing is required frequently to ensure microbiological safety in the manufacturing of food and beverages. IoT sensors can monitor important parameters like temperature and cleanliness, preventing contamination and preserving product quality.

*Data security and privacy:* Strong security measures are needed while handling sensitive microbiological data to prevent unwanted access or data breaches. Encryption, secure

communication protocols, and adherence to pertinent data protection laws should all be used in IoT solutions.

IoT and associated technologies, which allow real-time data collecting, automation, remote monitoring, and improved collaboration, present intriguing answers to these problems. However, data integrity, validation of IoT-generated data, and efficient integration with current laboratory operations should all be taken into account when implementing the IoT in microbiology.

## 5. Proposed System

In this section, the major emerging technologies used for the proposed system are presented, in addition to the architecture of our proposed system.

### 5.1. Involved Technologies

The following cutting-edge technologies, when combined to form our suggested system, can provide a strong system with enhanced virus detection capabilities.

### 5.1.1. Internet of Things (IoT)

The IoT is a cutting-edge new technology that makes it possible to connect any commonly used physical devices to the internet. This fact creates a vast global network of distinctive items that can communicate with one another to fulfill predetermined tasks, resulting in a variety of positive effects in various scientific fields, including medicine [4]. Surprisingly, wearable IoT-based gadgets can be used in healthcare and have applications, offering a wide spectrum of new possibilities because of pervasive connectivity [5].

Moreover, a secure monitoring system for IoT healthcare data on the cloud is made possible thanks in large part to the Internet of Things (IoT). By connecting various gadgets and sensors to the internet, IoT technology enables data collection, transmission, and exchange. IoT devices can include wearable health monitors, smart medical equipment, remote patient monitoring systems, and more in the context of healthcare. Some major operations that directly relate the IoT to the healthcare sector are the following [4,5,8]:

- *Data collection and monitoring:* IoT devices in the healthcare industry are able to continuously collect patient health information, including vital signs like heart rate, blood pressure, temperature, and glucose levels. These gadgets send the information to the cloud for centralized storage and supervision.
- *Real-time data streaming:* Real-time data streaming is frequently provided by IoT devices, allowing healthcare providers and medical experts to view and track patients' health conditions remotely. For the purpose of identifying anomalies and making any necessary interventions in a timely manner, real-time monitoring is essential.
- *Cloud-based data storage:* To securely store the collected data, IoT healthcare devices often rely on cloud-based storage options. In order to ensure that patient health data is available from any location while protecting data integrity and confidentiality, cloud storage enables scalable and cost-effective data management.
- *Secure data transmission:* It is critical to use secure communication protocols (like SSL/TLS) to encrypt data when it is being transported from IoT devices to the cloud. This guarantees data confidentiality and avoids illegal interception.
- *Authentication and access control:* Strong authentication procedures are needed for a secure monitoring system for IoT healthcare data in order to confirm the legitimacy of users accessing the cloud system. Based on the user's role and rights, Role-Based Access Control (RBAC) can be used to limit data access.
- *Data encryption and decryption:* Using encryption methods, such as the previously described Fernet encryption, guarantees that private medical information is encrypted before being saved in the cloud. Only authorized users are permitted to decrypt files, protecting the secrecy of the data.
- *Monitoring and anomaly detection:* IoT devices and cloud systems can be fitted with monitoring and anomaly detection methods to detect any strange patterns or potential

security breaches in real time. As a result, the system can better react quickly to security problems.

- *Regulatory compliance:* Cloud-based IoT healthcare systems must abide by all applicable rules, including the GDPR, HIPAA, and other data protection legislation. Compliance guarantees patient confidentiality and the safe handling of medical data.
- *Data backup and redundancy:* In order to guarantee data accessibility and disaster recovery in the event of hardware breakdowns or system outages, cloud-based IoT systems can make use of the redundancy and backup options offered by the cloud provider.

Healthcare professionals can monitor patients remotely, continuously measure health metrics, and provide individualized care while ensuring data security and privacy using IoT technologies. The healthcare sector is enabled to take advantage of real-time data insights, improve patient outcomes, and improve overall healthcare service delivery by fusing IoT devices with a secure monitoring system in the cloud.

### 5.1.2. Wireless Sensor Networks (WSN)

Wireless Sensor Networks (WSNs) are networks of small, inexpensive sensor nodes that use wireless communication to gather and send data from their surroundings. These sensor nodes can monitor physical or environmental variables because they are furnished with a variety of sensors, including temperature, humidity, light, motion, and gas sensors. The primary goal of WSN is to connect IoT-based devices (such as those belonging to a patient) in order to offer helpful information whenever and wherever it is required, such as in a hospital [4].

WSNs have a number of benefits over conventional wired sensor networks, including quick deployment, adaptability, scalability, and affordability. Numerous industries, such as environmental monitoring, industrial automation, healthcare, agriculture, transportation, and smart cities, all find use for them. WSNs can increase their capabilities and potential applications by merging them with other internet technologies [15].

A wide range of opportunities for improving data gathering, analysis, and decision-making capabilities are made possible by the integration of WSNs with other internet technologies. WSNs can aid in creating intelligent and interconnected systems in various sectors by utilizing the power of connection and cutting-edge computation [4].

### 5.1.3. Cloud Computing (CC)

The transmission of on-demand computing resources, such as storage, processing power, and software programs, through the internet is referred to as cloud computing. Cloud computing enables users to access and utilize these resources remotely from any location with an internet connection instead of relying on local servers or personal devices. The adoption of cloud computing in the healthcare industry has improved the effectiveness, usability, and security of healthcare services. Also, in the field of medicine, CC can give medical professionals like doctors and nurses access to patient records anytime and wherever they are [16].

The healthcare industry now has access to scalable infrastructure, data-sharing capabilities, cutting-edge analytics, and remote healthcare delivery thanks to the integration of cloud computing with other internet technologies [16]. Collaboration between healthcare providers is facilitated, data-driven decision-making is made possible, and ultimately, patient care and outcomes are improved.

### 5.1.4. Machine Learning (ML)

Machine learning (ML) algorithms can help identify various virus kinds more accurately so that they can be dealt with and prevented as soon as feasible. Additionally, this will stop susceptible populations from spreading. Based on the properties of the air in the environment, ML can be helpful in the prognosis, diagnosis, and control of patients infected with certain types of flu throughout the period of preventing and controlling the pandemic [20]. The respiratory pattern differs from the typical cold and other infections,

according to the most recent clinical studies [21]. However, with the improved efficiency of these algorithms and the capacity to utilize huge volumes of data with various properties, the advantages of adopting ML techniques in risk management will be highly beneficial [22]. Artificial Intelligence (AI) technology and machine learning can be treated as subversive technologies that change established procedures for monitoring effective virus recognition, and in particular, because of the current situation, it could contribute to the prevention and control of fatal viruses like COVID-19.

### *5.2. System's Primary Goals*

The primary goals of the proposed secure monitoring system for IoT healthcare data in the cloud are listed below:

1. *Confidentiality:* Protect the privacy and confidentiality of sensitive healthcare data, preventing unauthorized access or disclosure.
2. *Integrity:* Ensure the accuracy, consistency, and reliability of healthcare data by preventing unauthorized modification or tampering.
3. *Availability:* Maintain high availability of healthcare data and systems, minimizing downtime and ensuring that authorized users can access the data when needed.
4. *Compliance:* Meet the legal and regulatory requirements for protecting healthcare data, such as HIPAA, GDPR, or any other applicable regulations.
5. *Detection and response:* Detect and respond to security incidents, anomalies, or breaches in a timely manner to minimize the impact on patient safety and data security.
6. *Auditability:* Enable comprehensive logging and auditing capabilities to track and monitor user activities, system events, and data access for forensic analysis and compliance purposes.

The primary goals of the proposed system are depicted in Figure 1. By focusing on these implementation aspects and goals, the proposed secure monitoring system for IoT healthcare data in the cloud can help safeguard sensitive healthcare information, maintain the trust of patients and healthcare providers, and ensure the delivery of quality healthcare services.

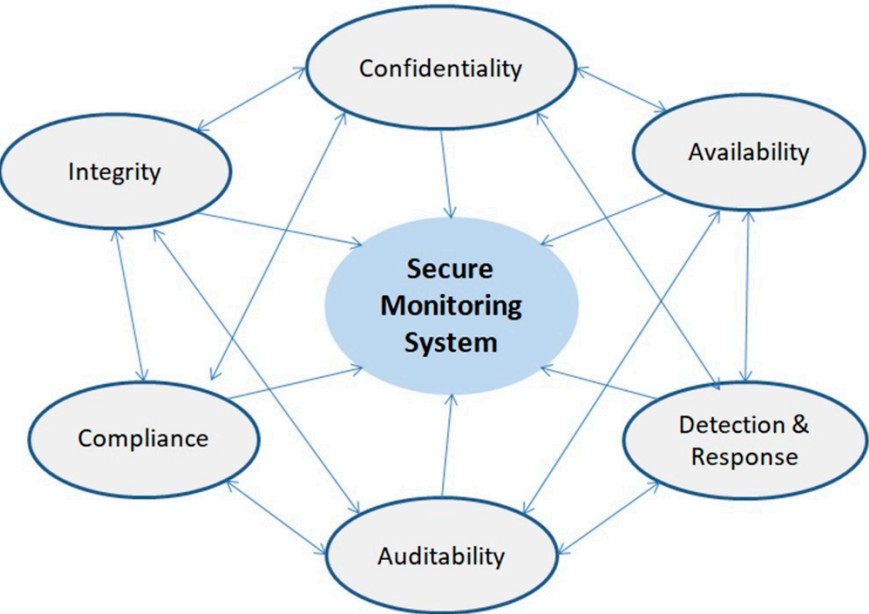

**Figure 1.** The primary goals of the proposed system.

### *5.3. System Architecture*

The main goal of this work is to introduce a secure machine learning monitoring system for IoT healthcare data in the cloud, which integrates the benefits delivered by a

machine learning scenario and the immediacy of the data produced by the IoMT with the computation power offered from a cloud computing server, operated through a wireless network. This system could produce, manage, and control IoMT data produced by air condition-quality sensors and thermal-IR cameras, in addition to various mobile devices.

The XMPP (Extensible Messaging and Presence Protocol) has been used to communicate the data generated by various devices, such as sensors and cameras. This protocol has undergone extensive use and testing and is an open standard. It is based on an efficient model, the publish/subscribe model presented in Figure 2 [23].

## Publish/Subscribe Communication Model

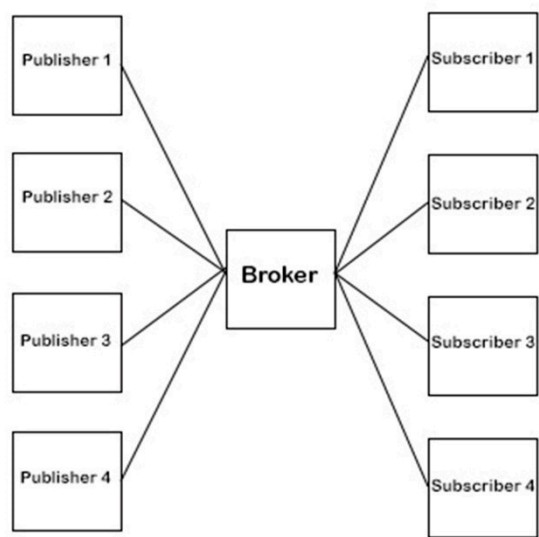

**Figure 2.** Publish/subscribe communication model.

In this paradigm, data is sent from a publisher (an IoT device) to a broker device in a safe and effective manner. Any subscriber device that requests a certain "topic" is then forwarded the updated data by the broker device. As an illustration, the temperature sensor will publish temperature information to the broker under the "temperature" subject. Those requesting the temperature topic will have access to these data. The requirements of storage and cloud systems are met by the XMPP protocol. Additionally, a framework that provided an abstraction of the network devices was employed to create the system and facilitate communication across all devices, servers, and databases. The interface between the IoT devices [XMPP] and the API (Application Programming Interface), which is hosted in the user's device, is made simpler by this framework.

The system architecture for our suggested model implementation is depicted in Figure 3. Moreover, the operation of the proposed system in Figure 3 is analyzed below. Each surveillance infrastructure is under the direction of the cloud server. An installed infrastructure that is not directly connected to a cloud server may also provide data to it. To export the appropriate conclusions for each monitoring region, the cloud server managed and organized all the data that the system had acquired. Additionally, the cloud offers a safe environment for gathering, managing, and processing the data generated by the IoMT. The peripheral equipment of the system is connected either to an autonomous network and through it to the internet and from there to the proposed system or is directly connected to the network of the proposed system.

Thermal-IR cameras measure the ambient temperature and, more precisely, the temperature of people arriving or moving through the surveillance area. Each camera sends the pictures it has taken to a cloud server over a wireless network. A server in the cloud runs a data analysis algorithm that determines whether the temperature reading of each camera is greater than 37 Celsius. Then, to stop the spread and isolate that person in another,

separated space, a notification signal appears in the monitoring system of that area if a temperature over that particular threshold is observed.

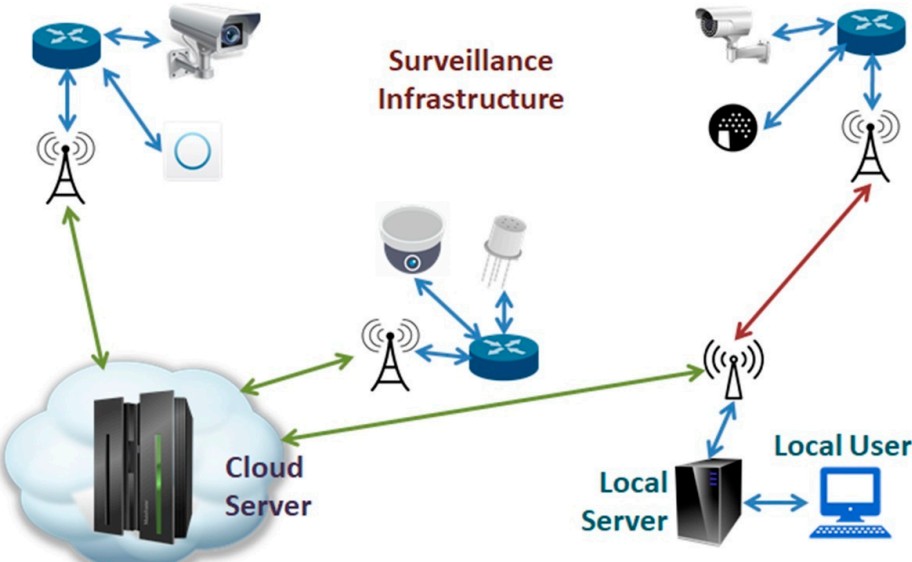

**Figure 3.** Proposed system's architecture.

The area's environment has a high level of air pollution, according to air quality sensors. The cloud server received the detection data via the wireless network. The data analysis algorithm established on the cloud server inspects if the rate of air pollution in the area exceeds normal levels. If it exceeds, then a notification signal appears in the monitoring system of that area, and then the local surveillance and control personnel will be notified. The entire proposed system is based on the algorithm proposed below, and it involves all existing encryption/decryption models to create a new model that will provide direct data transmission and management in a secure cloud environment. The operation of the proposed system based on the person's temperature and/or the overall air quality is depicted in Figure 4.

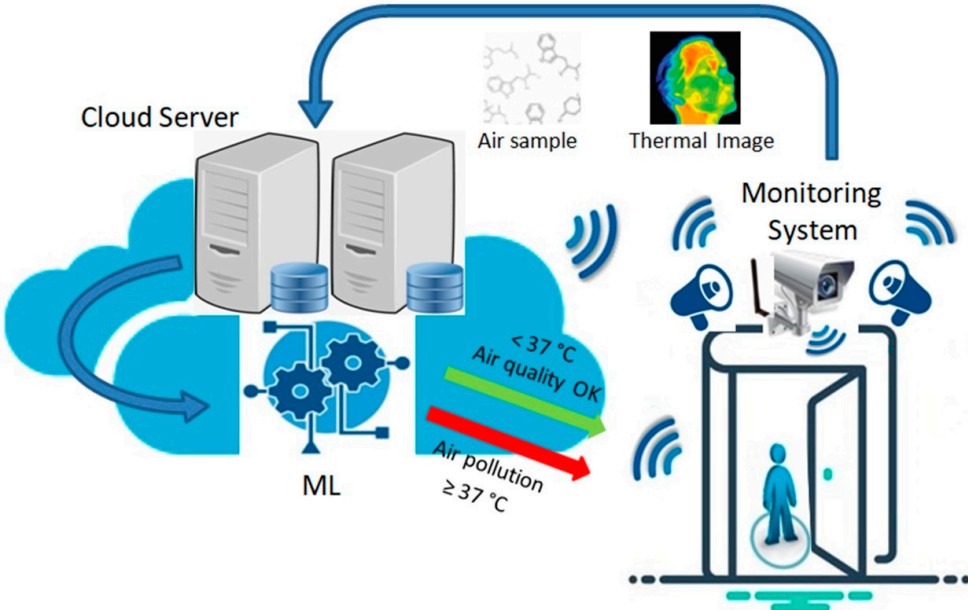

**Figure 4.** Operation of the proposed system based on the person's temperature and/or the overall air quality.

A machine learning scenario is set up in the cloud server that will be able to detect and inform about the state of the monitored area. After many calculations and iterations, the ML algorithm will be able to export automated conclusions about whether or not a location is at risk. It manages and controls the data shared from the IoT sensors, both cameras and air sensors, as well as the data produced and collected by the connected mobile devices. It will, therefore, alert the local surveillance and control people based on the data given via the IoT sensors. The goal of the ML algorithm is to be able to detect without the need for human supervision and then properly notify about the detection it made.

### 5.4. System's Algorithm Approach

The proposed system's algorithm is presented below with a Python source code implementation paradigm (Algorithm 1):

---

**Algorithm 1.** Secure IoT Healthcare Data in Cloud

---

```python
import ssl
import hashlib
import socket
import cryptography
from cryptography.fernet import Fernet
# Example IoT device data
iot_data = {
"patient_id": "123456",
"heart_rate": 75,
"temperature": 37.2
}
# Generate encryption key
encryption_key = Fernet.generate_key()
cipher_suite = Fernet(encryption_key)
# Encrypt IoT data
encrypted_data = cipher_suite.encrypt(str(iot_data).encode())
# Secure communication with SSL/TLS
context = ssl.create_default_context(ssl.Purpose.related-client)
context.check_hostname = False
context.verify_mode = ssl.CERT_NONE
# Connect to cloud server securely
cloud_server = "cloud server's host"
cloud_port = 443
cloud_socket = context.wrap_socket(socket.socket(socket.AF_INET),
server_hostname=cloud_server)
cloud_socket.connect((cloud_server, cloud_port))
# Send encrypted data to the cloud server
cloud_socket.sendall(encrypted_data)
# Receive response from cloud server
response = cloud_socket.recv(1024)
# Decrypt and validate the response
decrypted_response = cipher_suite.decrypt(response)
if hashlib.md5(response).hexdigest() == hashlib.md5(decrypted_response).hexdigest():
print("Response integrity verified: ", decrypted_response)
else:
print("Response tampering detected!")
# Close the connection
cloud_socket.close()
```

---

This snippet of code shows how a secure monitoring system for IoT healthcare data may be implemented simply in the cloud. Therefore, the proposed code might not cover every element of a system used in production; thus, we will try to improve all the aspects of our proposed system through future work. Furthermore, we can clarify each element

of it: *Import libraries*: The code begins by importing necessary libraries, including 'ssl', 'hashlib', 'Fernet' from 'cryptography.fernet', and 'socket'. *IoT device data*: A sample IoT device data dictionary ('iot_data') is created, simulating health-related information like patient ID, heart rate, and temperature. *Encryption key generation*: An encryption key is generated using Fernet from the 'cryptography.fernet' library. This key will be used for data encryption and decryption. *Data encryption*: The 'iot_data' dictionary is converted to a string and encrypted using the generated encryption key, resulting in 'encrypted_data'. *Secure communication setup*: An SSL/TLS secure communication context is created using the 'ssl' library, ensuring secure communication with the cloud server. Connect to cloud server: A secure connection is established with the cloud server using the given 'cloud_server' address and port number. *Send encrypted data*: The encrypted IoT data is sent to the cloud server using a secure connection. *Receive and validate response*: The code receives a response from the cloud server. The response is decrypted using the same encryption key. The code then compares the MD5 hash of the original response and the decrypted response to verify data integrity. *Print result*: Based on the validation, the code prints whether the response integrity is verified or if tampering is detected. *Close connection*: The secure connection to the cloud server is closed.

Moreover, a breakdown of the proposed algorithm is briefly described here: (1) Regarding the imported libraries, *ssl*: Provides support for secure sockets using the SSL/TLS protocols. *hashlib*: Offers hash functions, which are used for integrity validation. *socket*: Enables communication over sockets. *Cryptography*: A library for secure communication. *cryptography.fernet.Fernet*: A part of the cryptography library used for symmetric encryption (same key for encryption and decryption). (2) Regarding the IoT device data, a sample set of IoT device data ('*iot_data*') is created, representing health-related information like patient ID, heart rate, and temperature. (3) Regarding the generate encryption key, an encryption key is generated using the Fernet symmetric encryption scheme from the "*cryptography*" library. (4) Regarding the encrypt IoT data, the IoT data is converted to a string, encoded, and then encrypted using the generated encryption key ('*cipher_suite*'). The result is '*encrypted_data*'. (5) Regarding the secure communication setup, an SSL/TLS secure communication context (context) is created. The purpose is set to "*ssl.Purpose.related-client*" (which might be a typo and could be "*ssl.Purpose.SERVER_AUTH*"), and certain verification checks are disabled to simplify the example. (6) Regarding the connect to cloud server securely, a secure socket is established ('*cloud_socket*') with the cloud server at a specified address ('*cloud_server*') and port ('*cloud_port*'). The SSL/TLS wrap_socket method is used to secure the communication. (7) Regarding the send encrypted data to cloud server, the encrypted IoT data is sent to the cloud server over the secure connection. (8) Regarding the receive response from cloud server, the script waits to receive a response from the cloud server. (9) Regarding the decrypt and validate response, the received response is decrypted using the same encryption key. Moreover, the integrity of the response is then validated by comparing the MD5 hash of the original response and the decrypted response. If the hashes match, the response integrity is considered verified; otherwise, a tampering detection message is printed. (10) Regarding closing the connection, the secure connection to the cloud server is closed. Particularly, this algorithm is a basic illustration of secure communication practices, including data encryption and integrity verification, commonly used in IoT applications where maintaining the confidentiality and integrity of the transmitted data is crucial.

The algorithm demonstrated in this Python source code scenario is a simplified example crafted for illustrative purposes, aiming to cover essential security considerations of a real-world implementation. For a comprehensive solution, additional measures such as robust error handling, advanced authentication mechanisms, secure storage practices, and compliance with relevant standards and regulations should be implemented.

This proposed source code algorithm illustrates the encryption of IoT data using the *cryptography* library and the utilization of SSL/TLS to establish a secure connection with the cloud server. Additionally, it incorporates a basic integrity check using the *hashlib* library to verify the response received from the server.

Table 1 presents the capabilities of our proposed system compared to other related systems. As it is clearly shown, our system can detect not only COVID-19 and other infectious diseases but also air pollution levels in different environments, e.g., rooms, halls, etc. Furthermore, our monitoring system provides security and privacy for its users since it integrates many security features compared to other proposed schemes.

**Table 1.** Proposed system compared to other related systems.

| Research Work | Integrated Technologies | Framework System | Detection | Network-Based | Security | Privacy | Real-Time Data Management |
|---|---|---|---|---|---|---|---|
| [3] | Cloud, Internet of Things, wireless network | nCapp | COVID-19 diagnosis | Yes | N/A | N/A | Yes |
| [4] | Smartphones, AI, cameras, microphones, multiple sensors, Internet of Things, WSNs | Smartphone-based | COVID-19 detection | Yes | Yes | Yes | Yes |
| [5] | ATMEGA328, ESP8266-12F, mobile app, humidity, and temperature sensors | AC/Humidifier IoT Module | Infectious disease control | Yes | N/A | N/A | Yes |
| [7] | Cloud computing, edge computing, fog computing, Internet of Health Things | Based on fog computing for Internet of Health Things (Fog-IoHT) applications | N/A | Yes | N/A | N/A | Yes |
| **Proposal** | Cloud computing, machine learning, AI, IoT | Secure IoT-based Monitoring System | COVID-19 detection, fever detection, diagnosis of other infectious diseases, indication of air pollution level in rooms | Yes | Yes | Yes | Yes |

## 6. Application Fields

Public buildings will be the primary target of this implementation system because they are places where a lot of people congregate and have high traffic volumes. Nevertheless, in the early stages of this model, we are not so much interested in data security, as the main purpose is immediacy and speed in forecasting and informing about possible risk detection.

The detection of deadly viruses will benefit greatly from the usage of our suggested model. The proposed method can be used to detect possible viruses that are in the air or on objects and surfaces in both indoor and outdoor settings. Moreover, the proposed system can be used to check incoming people if they are potentially infected by a virus, warning them appropriately.

As a result, it is true that our suggested model can be used in medical facilities, hospitals, universities, amusement parks, and outdoor venues, offering a practical means of fending off not only well-known viruses but also novel spreading viruses like the recently discovered deadly Coronavirus, COVID-19.

Evidently, two industries that suffered during the pandemic are airlines and tourism. One of the first actions to minimize the impact of spreading a virus is to stay isolated as much as possible and, of course, not to travel abroad. Thus, the totals of airline companies are obligated to cancel the majority of their scheduled itineraries as a precaution against spreading the disease. Consequently, countries that are considered touristic may face a huge financial recession. Thus, the deployment of the proposed system could mitigate the losses of both industries.

### IoT in Healthcare Application Fields

IoT and other new technologies have a variety of uses in the healthcare industry that have the potential to completely change how care is delivered [9,12]. The following are some important application areas where these technologies can have a big impact [11,13]:

*Remote patient monitoring*: IoT devices allow for real-time data transmission to the cloud and continuous monitoring of a patient's vital signs and health parameters. This is especially helpful for older patients, post-operative treatment, and controlling chronic illnesses.

*Telemedicine and virtual health*: IoT-driven telemedicine technologies make it easier to schedule follow-up appointments, remote diagnostics, and virtual consultations while maintaining data security.

*Personalized medicine*: IoT devices gather patient-specific health data, which, when combined with cloud-based analytics, allows healthcare practitioners to customize interventions and treatments based on the needs of individual patients.

*Disease management and prevention*: IoT devices can help manage treatment regimens and track the evolution of diseases. Anomalies that are discovered early can assist in averting complications and advance preventive healthcare.

*Pharmaceutical research and manufacturing*: IoT sensors improve quality control and regulatory compliance while ensuring ideal circumstances for medication development and manufacture.

*Clinical trials*: The use of IoT devices increases the dependability of trial results and ensures the security of sensitive patient data by collecting precise, ongoing data from trial participants.

*Elderly care*: Senior living facilities that use IoT-based monitoring systems encourage resident safety and well-being while securely storing data in the cloud for caregiver access.

*Health and wellness wearables*: IoT devices monitor physical activity, sleep habits, and general well-being, enabling people to actively control their health.

*Emergency response and disaster management*: IoT-enabled medical equipment enables healthcare professionals to remotely assess patient conditions during emergencies and natural catastrophes, facilitating quick response.

*Public health surveillance*: IoT data gathering and analytics help to monitor population health trends, track disease outbreaks, and create efficient public health interventions.

*Data-driven decision-making*: When paired with cloud-based analytics and artificial intelligence, IoT-generated data helps healthcare managers and providers make decisions based on the best available evidence.

*Healthcare facility management*: IoT sensors track the health of medical equipment, the environment, and patient flow, enhancing both operational effectiveness and patient care.

*Precision agriculture and nutrition*: IoT devices keep an eye on crop health and soil conditions to help produce foods that are nutrient-rich and encourage preventive healthcare.

*Medical imaging and diagnostics*: Medical imaging equipment that is IoT-connected streamlines data collection and transfer, increasing the precision and speed of diagnostic procedures.

*Medical supply chain management*: IoT sensors in the supply chain monitor the state of the equipment and medicines to ensure quality and avoid shortages.

*Medical training and simulation*: Medical education and skill development are improved by IoT-driven simulation systems that provide realistic training experiences.

*Data privacy and security*: Blockchain and other emerging technologies can improve the security and privacy of patient data, guaranteeing compliance with rules like HIPAA and GDPR.

IoT, emerging technologies, and secure cloud-based solutions have the power to revolutionize healthcare delivery, enhance patient outcomes, increase operational effectiveness, and spur innovation across a range of healthcare-related industries.

## 7. Conclusions and Future Work

Emerging technologies should be incorporated into healthcare systems' plans since they can be beneficial in a number of ways. While fewer individuals are physically crammed into hospital facilities, patients can nevertheless receive the same level of clinical care. Additionally, using various AI-based triage systems could decrease the clinical burden on doctors. An online medical system may instruct individuals on the value of hand cleanliness, assist patients in identifying early symptoms, and refer them for medical care

should symptoms escalate. In conclusion, a wide range of technologies are now available that can be utilized to supplement and improve existing public health methods, even if the COVID-19 pandemic is still being addressed by traditional public health approaches around the world. The proposed system is a reliable model that, due to its cutting-edge and potent identification methods, might restrict virus dissemination.

To extract instantaneous data at the sensor level, an expansion of this model that adopts a federated learning scenario at the local node of each sensor is envisaged as future development. In order to improve the detecting situation, installing humidity sensors will be a helpful addition to the surveillance infrastructure.

**Author Contributions:** Conceptualization, C.L.S., A.P.P. and V.A.M.; methodology, C.L.S., A.P.P. and K.E.P.; software, C.L.S. and A.P.P.; validation, C.L.S., A.P.P., V.A.M., M.P.K. and K.E.P.; formal analysis, C.L.S. and K.E.P.; investigation, C.L.S. and A.P.P.; resources, C.L.S.; data curation, C.L.S. and A.P.P.; writing—original draft preparation, C.L.S.; writing—review and editing, C.L.S.; visualization, C.L.S. and A.P.P.; supervision, K.E.P.; project administration, K.E.P. All authors have read and agreed to the published version of the manuscript.

**Funding:** This research received no external funding.

**Institutional Review Board Statement:** Not applicable.

**Informed Consent Statement:** Not applicable.

**Data Availability Statement:** The data presented in this study are available on request from the corresponding author. The data are not publicly available due to privacy.

**Conflicts of Interest:** The authors declare no conflict of interest.

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
