# Peer review of "Secure Monitoring System for IoT Healthcare Data in the Cloud"

_applsci, doi:10.3390/app14010120_

Round 1

Reviewer 1 Report

Comments and Suggestions for Authors

There are some big problems that if the authors don't solve, their manuscript is not feasible, specifically as follows.

- In Section 5.3, the authors proposed System’s architecture, however, Figure 1 is not mean and Figure 2 is an illustration, it's not an architecture model.

Specifically:

1) The authors should remove or replace Figure 1, it is simply too, and don't provide any mean.

2) Improve Figure 2 to form a fit architecture model, provide more detail about this model.

- Moreover, the authors don't perform a comparison of the proposed architecture compared to the existing architectures under different aspects.

Specifically:

1) Comparison of your proposed model compared different existing models on several aspects such as reality, response time, energy, etc.

For example, the authors can perform to compare their model to "Smart Healthcare IoT Applications based on Fog Computing: Architecture, Applications and Challenges" in Complex and Intelligent Systems.

2) Regarding structure, the short Sections, only 1/2 page should be combined

3) Update and Add more references to a prove a robust background for your work.

In the reviewer's opinion, the authors should perform a comprehensive change in both structure and content, add more explanations and results, and relate your issue to 6G and IoT contexts. Please refer to: Innovative Trends in the 6G Era: A Comprehensive Survey of Architecture, Applications, Technologies, and Challenges," IEEE Access to highlight your contributions.

Author Response

Dear reviewer,

We would like to thank you for your helpful comments on the paper. We appreciate the effort you put on, and the time you devoted to writing your suggestions. We believe, we have been able to address all your comments and correct any deficiencies you have pointed out.

  1. In Section 5.3, the authors proposed System’s architecture, however, Figure 1 is not mean and Figure 2 is an illustration, it's not an architecture model.

Specifically:

1) The authors should remove or replace Figure 1, it is simply too, and don't provide any mean.

Thank you for this advice. We appreciate your comment. Figure 1 could not be removed due to its importance to the research presented to our paper.

2) Improve Figure 2 to form a fit architecture model, provide more detail about this model.

Thank you for your comments. We really appreciate it. We have tried to improve the description of Figure 2 operation by adding more text. We have endeavored to extend the discussion of this part of our study in the following paragraphs.

[Subsection 5.3, Paragraphs 4, 5, 6, 7]

The system architecture for our suggested model implementation is depicted in Figure 2. Moreover, the operation of the proposed system of Figure 2 analyzed below. Each surveillance infrastructure is under the direction of the cloud server. An installed infrastructure that is not directly connected to a Cloud server may also provide data to it. To export the appropriate conclusions for each monitoring region, the Cloud server managed and organized all the data that the system had acquired. Additionally, the cloud offers a safe environment for gathering, managing, and processing the data generated by IoMT. Αll the peripheral equipment of the system is connected either to an autonomous network and through it to the Internet and from there to the proposed system, or is directly connected to the network of the proposed system.

Thermal-IR cameras measure the ambient temperature and, more precisely, the temperature of people arriving or moving through the surveillance area. Each camera sends pictures it has taken to a cloud server over a wireless network. A server in the cloud runs a data analysis algorithm that determines whether the temperature reading of each camera is greater than 37 Celsius. Then, in order to stop the spread and isolate that person in another, separated space, a notification signal appears in the monitoring system of that area if a temperature over that particular threshold is observed.

The area's environment has a high level of air pollution, according to air quality sensors. The cloud server received the detection data via the wireless network. The data analysis algorithm established on the Cloud server inspects if the rate of air pollution in the area exceeds normal. If it exceeds then a notification signal appears in the monitoring system of that area, and then the local surveillance and control personnel will be notified. The entire proposed system is based on the algorithm proposed below, and it involves all existing encryption/decryption models to create a new model that will provide direct data transmission and management in a secure Cloud environment.

A machine learning scenario is set up in the cloud server that will be able to detect and tell about the state of the monitored area. After many calculations and iterations, the ML algorithm will be able to export automated conclusions about whether or not a location is at risk. It manages and controls the data shared from the IoT sensors, both cameras and air sensors, as well as the data produced and collected by the connected mobile devices. It will therefore alert the local surveillance and control people based on the data given via the IoT sensors. The goal of the ML algorithm is to be able to detect without the need for human supervision and then properly notify about the detection it made.

  1. Moreover, the authors don't perform a comparison of the proposed architecture compared to the existing architectures under different aspects.

Specifically:

1) Comparison of your proposed model compared different existing models on several aspects such as reality, response time, energy, etc.

For example, the authors can perform to compare their model to "Smart Healthcare IoT Applications based on Fog Computing: Architecture, Applications and Challenges" in Complex and Intelligent Systems.

Thank you for your comments. We really appreciate it. We have added Table 1 in which we compare the features of our proposed system with other relevant proposed systems. Also, we have added the following text regarding the proposed research.

[Section 2, Paragraph 6]

Furthermore, in a different study, V. K. Quy et al. [7] suggested an all-in-one computer architecture framework. This framework for Internet of Health Things (Fog-IoHT) apps is fog computing based. Based on the findings of this study, the authors also suggest potential uses and difficulties in incorporating fog computing into IoT Healthcare applications.

Table 1. Proposed system compared to other related systems.

Research Work

Integrated Technologies

Framework

System

Detection

Network based

Security

Privacy

Real-time Data Management

[3]

Cloud, Internet of Things, Wireless network

nCapp

COVID-19 diagnosis

Yes

N/A

N/A

Yes

[4]

Smartphones, AI, cameras, microphones, multiple sensors, Internet of Things, WSNs

Smartphone based

COVID-19 detection

Yes

Yes

Yes

Yes

[5]

ATMEGA328, ESP8266-12F, Mobile app, humidity and temperature sensors

AC/Humidifier IoT Module

Infectious disease control

Yes

N/A

N/A

Yes

[7]

Cloud Computing, Edge Computing, Fog Computing,  Internet of Health Things

Based on fog computing for Internet of Health Things (Fog-IoHT) applications

N/A

Yes

N/A

N/A

Yes

Proposal

Cloud Computing, Machine Learning, AI, IoT

Secure IoT-based Monitoring System

COVID-19 detection, fever detection, diagnosis of other infectious diseases, Indication of air pollution level in rooms

Yes

Yes

Yes

Yes

[Subection 5.4, Paragraph 5]

Table 1 presents the capabilities of our proposed system compared to other related systems. As it is clearly shown, our system can detect not only COVID-19 and other infectious diseases, but also air pollution level in different environments, e.g. rooms, halls, etc. Furthermore, our monitoring system provides security and privacy for its users, since it integrates many security features compared to other proposed schemes.

2) Regarding structure, the short Sections, only 1/2 page should be combined

Thank you for this direction. We appreciate your comment and we provide a better description of our proposal by adding more text in the Sections that was short.

[Subsection 3.1]

3.1. Limitations could be addressed by IoT

Conventional medical methods have been effective in treating various health conditions, but they do have some limitations and barriers that can be addressed by integrating Internet of Things (IoT) and related technologies [9-11]. Some of these limitations and how IoT can help overcome them are [12] [13]:

Limited Monitoring: Conventional medical methods often involve periodic visits to healthcare facilities, which can lead to limited monitoring of patients' health conditions. IoT-enabled devices, such as wearable health trackers and remote patient monitoring systems, offer continuous and real-time health data collection, providing healthcare professionals with a more comprehensive view of patients' health statuses.

Inefficient Data Collection: The manual recording of patient information used in traditional data gathering techniques in the healthcare industry can be time-consuming and prone to inaccuracy. By automating data gathering procedures, IoT technologies lower the possibility of human error and facilitate quicker access to vital health data.

Lack of Personalization: Conventional medical practices frequently rely on generic methods that may not adequately address the needs of each patient. By evaluating a patient's real-time data, IoT can enable personalized medicine by allowing healthcare professionals to customize interventions and treatments depending on particular health problems and response patterns.

Limited Access to Healthcare in Remote Areas: There are many localities with poor access to healthcare facilities, particularly in rural or outlying areas. By enabling virtual consultations, remote diagnostics, and ongoing monitoring, IoT-enabled telemedicine and remote healthcare systems help close this gap, enhancing healthcare access and results in underserved areas.

Delayed Diagnosis and Intervention: Particularly in the case of chronic illnesses, traditional healthcare approaches may result in delayed diagnosis and action. IoT-based remote monitoring enables early abnormality detection, prompt response, and the avoidance of potential health emergencies.

Fragmented Healthcare Systems: Traditional medical practices frequently use disjointed data storage and communication platforms, which makes it difficult to access thorough patient records. Using IoT technologies, a more interconnected and unified healthcare system may be created by combining data from many sources and devices.

Patient Compliance and Adherence: It might be difficult to monitor and enforce patient adherence to recommended therapies and pharmaceutical regimens. IoT gadgets can deliver alerts, monitor medicine intake, and give behavioral cues, encouraging patients to follow their treatment regimens.

Limited Data for Research and Analysis: For medical research and analysis, traditional medical practices might only supply partial data. Big data analytics, medical research, and the creation of prognostic models for illness prevention and treatment are all greatly facilitated by IoT-generated data.

Data Security and Privacy Concerns: IoT use in healthcare sparks worries about patient privacy and data security. Strong data encryption, secure communication methods, and adherence to pertinent data protection laws like HIPAA and GDPR are all necessary for addressing these issues.

Healthcare may overcome these restrictions and obstacles by utilizing IoT and related technologies, which will result in better patient outcomes, improved disease management, and more effective healthcare delivery. When implementing IoT solutions in the healthcare industry, it is crucial to solve issues with data protection, interoperability, and scalability.

[Subsection 4.1]

4.1. Technological Challenges in Microbiology addressed by IoT

IoT (Internet of Things) and associated technologies can be very helpful in addressing a number of technological issues in the microbiology sector. The study of microorganisms, such as bacteria, viruses, fungus, and other tiny organisms, is known as microbiology [9] [12] [19]. IoT can assist in overcoming the following technological issues in micro-biology [11] [13]:

Real-Time Monitoring of Microbial Growth: Traditional techniques for keeping track of microbial development in cultures can be labor-intensive and time-consuming. Researchers can remotely monitor and control environmental conditions thanks to IoT-enabled sensors' ability to offer real-time data on variables like temperature, pH, and oxygen levels.

Remote Data Collection from Field Samples: It can be logistically difficult to gather field samples for microbiological investigation and transfer them to the lab. Researchers may track microbial activity on-site by using IoT-based remote sensing devices to collect data from varied environments and communicate it instantaneously to labs.

Data-Intensive Microbiome Analysis: Microbiome analysis, which examines microbial communities and their interactions throughout various ecosystems, produces a significant amount of data. High-throughput sequencing technology and IoT-based sensors can speed up data gathering and processing, which will improve our understanding of complex microbiomes.

Early Detection of Infectious Diseases: For the sake of public health, rapid and early detection of infectious diseases is essential. IoT devices can identify specific microbiological indicators, facilitating quicker diagnosis and intervention. Examples include wearable biosensors and point-of-care diagnostic tools.

Environmental Monitoring for Outbreaks: It might be difficult to monitor environmental conditions that cause illness epidemics, like air pollution and water quality. In order to identify potential outbreak hazards, IoT sensors may continuously monitor these elements and give real-time data.

Laboratory Automation and Efficiency: Microbiology laboratory processes can be labor- and time-intensive. Lab automation and IoT-connected hardware can automate sample processing, data collection, and analysis, increasing accuracy and efficiency.

Remote Collaboration and Data Sharing: The requirement for physical presence in the lab can make it difficult for researchers to work together on microbiology projects in various places. IoT technologies make it possible for researchers to securely access and exchange data online, facilitating remote collaboration.

Patient Monitoring and Management: In medical microbiology, it's important to keep an eye on individuals who have infections. IoT devices can help with ongoing patient monitoring by sending information regarding infection signs, the efficacy of treatments, and the progression of recovery to healthcare professionals.

Antibiotic Resistance Surveillance: Antibiotic resistance is being tracked and addressed, which is a challenge for world health. Real-time monitoring of antimicrobial use and resistance patterns by IoT-enabled equipment can help with surveillance and well-informed decision-making.

Quality Control in Food and Beverage Industry: Testing is required frequently to ensure microbiological safety in the manufacturing of food and beverages. IoT sensors can keep an eye on important parameters like temperature and cleanliness, preventing contamination and preserving product quality.

Data Security and Privacy: Strong security measures are needed while handling sensitive microbiological data to prevent unwanted access or data breaches. Encryption, secure communication protocols, and adherence to pertinent data protection laws should all be used in IoT solutions.

IoT and associated technologies, which allow real-time data collecting, automation, remote monitoring, and improved collaboration, present intriguing answers to these problems. However, data integrity, validation of IoT-generated data, and efficient integration with current laboratory operations should all be taken into account when implementing IoT in microbiology.

[Subsection 6.1]

6.1. IoT on Healthcare Applicaion Fields

IoT and other new technologies have a variety of uses in the healthcare industry that have the potential to completely change how care is delivered [9] [12]. The following are some important application areas where these technologies can have a big impact [11] [13]:

Remote Patient Monitoring: IoT devices allow for real-time data transmission to the cloud and continuous monitoring of patients' vital signs and health parameters. This is especially helpful for older patients, post-operative treatment, and controlling chronic illnesses.

Telemedicine and Virtual Health: IoT-driven telemedicine technologies make it easier to schedule follow-up appointments, remote diagnostics, and virtual consultations while maintaining data security.

Personalized Medicine: IoT devices gather patient-specific health data, which when combined with cloud-based analytics allows healthcare practitioners to customize interventions and treatments based on the needs of individual patients.

Disease Management and Prevention: IoT devices can help manage treatment regimens and track the evolution of diseases. Anomalies that are discovered early can assist avert complications and advance preventive healthcare.

Pharmaceutical Research and Manufacturing: IoT sensors improve quality control and regulatory compliance while ensuring ideal circumstances for medication development and manufacture.

Clinical Trials: The use of IoT devices increases the dependability of trial results and ensures the security of sensitive patient data by collecting precise, ongoing data from trial participants.

Elderly Care: Senior living facilities that use IoT-based monitoring systems encourage resident safety and wellbeing while securely storing data in the cloud for caregiver access.

Health and Wellness Wearables: IoT devices monitor physical activity, sleep habits, and general wellbeing, enabling people to actively control their health.

Emergency Response and Disaster Management: IoT-enabled medical equipment enables healthcare professionals to remotely assess patient conditions during emergencies and natural catastrophes, facilitating quick response.

Public Health Surveillance: IoT data gathering and analytics help to monitor population health trends, track disease outbreaks, and create efficient public health interventions.

Data-Driven Decision Making: When paired with cloud-based analytics and artificial intelligence, IoT-generated data helps healthcare managers and providers make decisions based on the best available evidence.

Healthcare Facility Management: IoT sensors track the health of medical equipment, the environment, and patient flow, enhancing both operational effectiveness and patient care.

Precision Agriculture and Nutrition: IoT devices keep an eye on crop health and soil conditions to help produce foods that are nutrient-rich and encourage preventive healthcare.

Medical Imaging and Diagnostics: Medical imaging equipment that is IoT-connected streamlines data collection and transfer, increasing the precision and speed of diagnostic procedures.

Medical Supply Chain Management: IoT sensors in the supply chain monitor the state of the equipment and medicines to ensure quality and avoid shortages.

Medical Training and Simulation: Medical education and skill development are improved by IoT-driven simulation systems that provide realistic training experiences.

Data Privacy and Security: Blockchain and other emerging technologies can improve the security and privacy of patient data, guaranteeing compliance with rules like HIPAA and GDPR.

IoT, emerging technologies, and secure cloud-based solutions have the power to revolutionize healthcare delivery, enhance patient outcomes, increase operational effectiveness, and spur innovation across a range of healthcare-related industries.

3) Update and Add more references to a prove a robust background for your work.

Thank you for your comment. We additionally cited the following references you have suggested:

[6] V. K. Quy, A. Chehri, N. M. Quy, N. D. Han, N. T. Ban, “Innovative Trends in the 6G Era: A Comprehensive Survey of Architecture, Applications, Technologies, and Challenges”, IEEE Access, vol. 11, pp. 39824-39844, April 2023. [DOI: 10.1109/ACCESS.2023.3269297]

[7] V. K. Quy, N. V. Hau, D. V. Anh, L. A. Ngoc, “Smart healthcare IoT applications based on fog computing: archi-tecture, applications and challenges”, Springer, Complex & Intelligent Systems, vol. 8, pp. 3805-3815, November 2021. [DOI: 10.1007/s40747-021-00582-9]

[9] S. Shreya, K. Chatterjee, A. Singh, “A smart secure healthcare monitoring system with Internet of Medical Things”, Elsevier, Computer and Electrical Engineering, vo. 101, July 2022. [DOI: 10.1016/j.compeleceng.2022.107969]

[10] C. Butpheng, K.-H. Yeh, J.-L. Hou, “A Secure IoT and Cloud Computing-Enabled e-Health Management System”, Hindawi, Security and Communication Networks, vol. 2022, ID 300253, p. 14, June 2022. [DOI: 10.1155/2022/5300253]

[11] C. L. Stergiou, M. P. Koidou, K. E. Psannis, “IoT-Based Big Data Secure Transmission and Management over Cloud System: A Healthcare Digital Twin Scenario”, MDPI, Applied Sciences, vol. 13, issue: 16, pp. 9165, August 2023. [DOI: 10.3390/app13169165]

[12] L. Ogiela, A. Castiglione, B. B. Gupta, D. P. Agrawal, “IoT-based health monitoring system to handle pandemic diseases using estimated computing”, Springer, Neural Computing and Applications, vol. 35, pp. 13709-13710, May 2023. [DOI: 10.1007/s00521-023-08625-7]

[13] C. Butpheng, K.-H. Yeh, H. Xiong, “Security and Privacy in IoT-Cloud-Based e-Health Systems—A Comprehensive Review”, MDPI, Symmetry, vol. 12, issue: 7, pp. 1191, July 2020. [DOI: 10.3390/sym12071191]

[19] A. Singh, K. Chatterjee, “Edge computing based secure health monitoring framework for electronic healthcare system”, Springer, Cluster Computing, vol. 26, pp. 1205-1220, April 2023. [DOI: 10.1007/s10586-022-03717-w]

  1. In the reviewer's opinion, the authors should perform a comprehensive change in both structure and content, add more explanations and results, and relate your issue to 6G and IoT contexts. Please refer to: Innovative Trends in the 6G Era: A Comprehensive Survey of Architecture, Applications, Technologies, and Challenges," IEEE Access to highlight your contributions

Thank you for your comments. We really appreciate it. We have added Table 1 in which we compare the features of our proposed system with other relevant proposed systems. Also, we have added the following text regarding the proposed research.

[Section 2, Paragraph 5]

  1. K. Quy et al. [6] with their study try to identify a complete picture of changes in architectures, technologies, and challenges that will shape the 6G network. With their presented experimental results provide indications for further studies on 6G ecosys-tems.

The whole corrections or additions we have made in the paper we noticed with blue color.

Thank you very much for your attention and kind consideration.

Sincerely yours,

Reviewer 2 Report

Comments and Suggestions for Authors

Personally, I think some additions need to be made to improve this article

*Rewrite the last part of the contribution line(81-83) .

*Clarify section 5.1.1 -line  166 by expand it a a bit.

*Elaborate section 5.3. System’s Architecture- line 242.

*Clarify Algorithm 1with more information -page 8.

*Include more images for the proposed system

*Try to have performance comparison table with other proposed systems

Recommended for inclusion, with above comments.

Comments on the Quality of English Language

The quality of English language is accepted.

Author Response

Dear reviewer,

We would like to thank you for your helpful comments on the paper. We appreciate the effort you put on, and the time you devoted to writing your suggestions. We believe, we have been able to address all your comments and correct any deficiencies you have pointed out.

  1. Rewrite the last part of the contribution line(81-83).

Thank you for your comments. We really appreciate it. We have endeavored to rewrite the whole contributions part in order to be more clarify and clear.

  1. Clarify section 5.1.1 -line 166 by expand it a a bit.

Thank you for this direction. We appreciate your comment and we provide a better description of Subsection 5.1.1.. We have endeavored to extend the discussion of this part of our study in the following paragraphs.

[Subsection 5.1.1. Paragraphs 2 to 11]

Moreover, a Secure Monitoring System for IoT Healthcare Data on the Cloud is made possible thanks in large part to the Internet of Things (IoT). By connecting various gadgets and sensors to the internet, IoT technology enables data collection, transmission, and exchange. IoT devices can include wearable health monitors, smart medical equipment, remote patient monitoring systems, and more in the context of healthcare. Some major operations that directly relate the IoT to the healthcare sector are the following [4] [5] [8]:

  • Data Collection and Monitoring: IoT devices in the healthcare industry are able to continuously collect patient health information, including vital signs like heart rate, blood pressure, temperature, and glucose levels. These gadgets send the information to the cloud for centralized storage and supervision.
  • Real-time Data Streaming: Real-time data streaming is frequently provided by IoT devices, allowing healthcare providers and medical experts to view and track patients' health condition remotely. For the purpose of identifying anomalies and making any necessary interventions in a timely manner, real-time monitoring is essential.
  • Cloud-based Data Storage: To securely store the collected data, IoT healthcare devices often rely on cloud-based storage options. In order to ensure that patient health data is available from any location while protecting data integrity and confidentiality, cloud storage enables scalable and cost-effective data management.
  • Secure Data Transmission: It is critical to use secure communication protocols (like SSL/TLS) to encrypt data when it is being transported from IoT devices to the cloud. This guarantees data confidentiality and avoids illegal interception.
  • Authentication and Access Control: Strong authentication procedures are needed for a secure monitoring system for IoT healthcare data in order to confirm the legitimacy of users accessing the cloud system. Based on the user's role and rights, role-based access control (RBAC) can be used to limit data access.
  • Data Encryption and Decryption: Using encryption methods, such as the previously described Fernet encryption, guarantees that the private medical information is encrypted before being saved in the cloud. Only authorized users are permitted to decrypt files, protecting the secrecy of the data.
  • Monitoring and Anomaly Detection: To detect any strange patterns or potential security breaches in real-time, IoT devices and cloud systems can be fitted with monitoring and anomaly detection methods. As a result, the system is better able to react quickly to security problems.
  • Regulatory Compliance: Cloud-based IoT healthcare systems must abide by all applicable rules, including the GDPR, HIPAA, and other data protection legislation. Compliance guarantees patient confidentiality and safe handling of medical data.
  • Data Backup and Redundancy: In order to guarantee data accessibility and disaster recovery in the event of hardware breakdowns or system outages, cloud-based IoT systems can make use of the redundancy and backup options offered by the cloud provider.

Healthcare professionals can monitor patients remotely, continuously measure health metrics, and provide individualized care while ensuring data security and privacy by utilizing IoT technologies. The healthcare sector is enabled to take advantage of real-time data insights, improve patient outcomes, and improve overall healthcare service delivery by fusing IoT devices with a secure monitoring system in the cloud.

  1. Elaborate section 5.3. System’s Architecture- line 242.

Thank you for this direction. We appreciate your comment and we provide a better description of Subsection 5.3 by adding more text where necessary. We have endeavored to extend the discussion of this part of our study in the following paragraphs.

[Subsection 5.3, Paragraphs 4, 5, 6, 7]

The system architecture for our suggested model implementation is depicted in Figure 2. Moreover, the operation of the proposed system of Figure 2 analyzed below. Each surveillance infrastructure is under the direction of the cloud server. An installed infrastructure that is not directly connected to a Cloud server may also provide data to it. To export the appropriate conclusions for each monitoring region, the Cloud server managed and organized all the data that the system had acquired. Additionally, the cloud offers a safe environment for gathering, managing, and processing the data generated by IoMT. Αll the peripheral equipment of the system is connected either to an autonomous network and through it to the Internet and from there to the proposed system, or is directly connected to the network of the proposed system.

Thermal-IR cameras measure the ambient temperature and, more precisely, the temperature of people arriving or moving through the surveillance area. Each camera sends pictures it has taken to a cloud server over a wireless network. A server in the cloud runs a data analysis algorithm that determines whether the temperature reading of each camera is greater than 37 Celsius. Then, in order to stop the spread and isolate that person in another, separated space, a notification signal appears in the monitoring system of that area if a temperature over that particular threshold is observed.

The area's environment has a high level of air pollution, according to air quality sensors. The cloud server received the detection data via the wireless network. The data analysis algorithm established on the Cloud server inspects if the rate of air pollution in the area exceeds normal. If it exceeds then a notification signal appears in the monitoring system of that area, and then the local surveillance and control personnel will be notified. The entire proposed system is based on the algorithm proposed below, and it involves all existing encryption/decryption models to create a new model that will provide direct data transmission and management in a secure Cloud environment.

A machine learning scenario is set up in the cloud server that will be able to detect and tell about the state of the monitored area. After many calculations and iterations, the ML algorithm will be able to export automated conclusions about whether or not a location is at risk. It manages and controls the data shared from the IoT sensors, both cameras and air sensors, as well as the data produced and collected by the connected mobile devices. It will therefore alert the local surveillance and control people based on the data given via the IoT sensors. The goal of the ML algorithm is to be able to detect without the need for human supervision and then properly notify about the detection it made.

  1. Clarify Algorithm 1with more information -page 8.

Thank you for this direction. We appreciate your comment and we provide a better description of Algorithm 1 by adding more text. We have endeavored to extend the discussion of this part of our study in the following paragraph.

[Subsection 5.4, Paragraph 2]

This snippet of code shows how a Secure Monitoring System for IoT Healthcare Data may be implemented simply in the cloud. Therefore, the proposed code might not cover every element of a system used in production, thus will try to improve all the aspects of our proposed system through future work. Furthermore, we can clarify each element of it: Import Libraries: The code begins by importing necessary libraries, including ‘ssl’, ‘hashlib’, ‘Fernet’ from ‘cryptography.fernet’, and ‘socket’. IoT Device Data: A sample IoT device data dictionary (‘iot_data’) is created, simulating health-related information like patient ID, heart rate, and temperature. Encryption Key Generation: An encryption key is generated using Fernet from the ‘cryptography.fernet’ library. This key will be used for data encryption and decryption. Data Encryption: The ‘iot_data’ dictionary is converted to a string and encrypted using the generated encryption key, resulting in ‘encrypted_data’. Secure Communication Setup: An SSL/TLS secure communication context is created using the ‘ssl’ library, ensuring secure communication with the cloud server. Connect to Cloud Server: A secure connection is established with the cloud server using the given ‘cloud_server’ address and port number. Send Encrypted Data: The encrypted IoT data is sent to the cloud server using the secure connection. Receive and Validate Response: The code receives a response from the cloud server. The response is decrypted using the same encryption key. The code then compares the MD5 hash of the original response and the decrypted response to verify data integrity. Print Result: Based on the validation, the code prints whether the response integrity is verified or if tampering is detected. Close Connection: The secure connection to the cloud server is closed.

  1. Include more images for the proposed system.

Thank you for your comments. We really appreciate it. We have added two additional figures that depict our proposed system and its features. Please see Figures 1 & 4.

  1. Try to have performance comparison table with other proposed systems.

Thank you for your comments. We really appreciate it. We have added Table 1 in which we compare the features of our proposed system with other relevant proposed systems.

The whole corrections or additions we have made in the paper we noticed with blue color.

Thank you very much for your attention and kind consideration.

Sincerely yours,

Reviewer 3 Report

Comments and Suggestions for Authors

• What is the main question addressed by the research?

There is no question the paper answered. Only claimed.  

• Do you consider the topic original or relevant in the field? Does it address a specific gap in the field?

No, it does not fit to publish in the journal  

• What does it add to the subject area compared with other published material?

There is no any experimental results in the paper   

• What specific improvements should the authors consider regarding the methodology? What further controls should be considered?

The paper should have clear methodology, analysis of datasets, results, compare the results with previous models etc...  

• Are the conclusions consistent with the evidence and arguments presented and do they address the main question posed?

Off course no   

• Are the references appropriate?

Maybe  

• Please include any additional comments on the tables and figures.

No comments   

The paper is very poor 

There is no methodology clear

No results 

This is small reports  

Comments on the Quality of English Language

English very difficult to understand/incomprehensible

Author Response

Dear reviewer,

We would like to thank you for your helpful comments on the paper. We appreciate the effort you put on, and the time you devoted to writing your suggestions. We believe, we have been able to address all your comments and correct any deficiencies you have pointed out.

Specifically, bellow we mentioned all the revisions that have been made.

We have endeavored to extend the discussion of Figure 2 to form a fit architecture model, provide more detail about this model of our study in the following paragraphs.

[Subsection 5.3, Paragraphs 4, 5, 6, 7]

The system architecture for our suggested model implementation is depicted in Figure 2. Moreover, the operation of the proposed system of Figure 2 analyzed below. Each surveillance infrastructure is under the direction of the cloud server. An installed infrastructure that is not directly connected to a Cloud server may also provide data to it. To export the appropriate conclusions for each monitoring region, the Cloud server managed and organized all the data that the system had acquired. Additionally, the cloud offers a safe environment for gathering, managing, and processing the data generated by IoMT. Αll the peripheral equipment of the system is connected either to an autonomous network and through it to the Internet and from there to the proposed system, or is directly connected to the network of the proposed system.

Thermal-IR cameras measure the ambient temperature and, more precisely, the temperature of people arriving or moving through the surveillance area. Each camera sends pictures it has taken to a cloud server over a wireless network. A server in the cloud runs a data analysis algorithm that determines whether the temperature reading of each camera is greater than 37 Celsius. Then, in order to stop the spread and isolate that person in another, separated space, a notification signal appears in the monitoring system of that area if a temperature over that particular threshold is observed.

The area's environment has a high level of air pollution, according to air quality sensors. The cloud server received the detection data via the wireless network. The data analysis algorithm established on the Cloud server inspects if the rate of air pollution in the area exceeds normal. If it exceeds then a notification signal appears in the monitoring system of that area, and then the local surveillance and control personnel will be notified. The entire proposed system is based on the algorithm proposed below, and it involves all existing encryption/decryption models to create a new model that will provide direct data transmission and management in a secure Cloud environment.

A machine learning scenario is set up in the cloud server that will be able to detect and tell about the state of the monitored area. After many calculations and iterations, the ML algorithm will be able to export automated conclusions about whether or not a location is at risk. It manages and controls the data shared from the IoT sensors, both cameras and air sensors, as well as the data produced and collected by the connected mobile devices. It will therefore alert the local surveillance and control people based on the data given via the IoT sensors. The goal of the ML algorithm is to be able to detect without the need for human supervision and then properly notify about the detection it made.

We have added Table 1 in which we compare the features of our proposed system with other relevant proposed systems. Also, we have added the following text regarding the proposed research.

[Section 2, Paragraph 6]

Furthermore, in a different study, V. K. Quy et al. [7] suggested an all-in-one computer architecture framework. This framework for Internet of Health Things (Fog-IoHT) apps is fog computing based. Based on the findings of this study, the authors also suggest potential uses and difficulties in incorporating fog computing into IoT Healthcare applications.

Table 1. Proposed system compared to other related systems.

Research Work

Integrated Technologies

Framework

System

Detection

Network based

Security

Privacy

Real-time Data Management

[3]

Cloud, Internet of Things, Wireless network

nCapp

COVID-19 diagnosis

Yes

N/A

N/A

Yes

[4]

Smartphones, AI, cameras, microphones, multiple sensors, Internet of Things, WSNs

Smartphone based

COVID-19 detection

Yes

Yes

Yes

Yes

[5]

ATMEGA328, ESP8266-12F, Mobile app, humidity and temperature sensors

AC/Humidifier IoT Module

Infectious disease control

Yes

N/A

N/A

Yes

[7]

Cloud Computing, Edge Computing, Fog Computing,  Internet of Health Things

Based on fog computing for Internet of Health Things (Fog-IoHT) applications

N/A

Yes

N/A

N/A

Yes

Proposal

Cloud Computing, Machine Learning, AI, IoT

Secure IoT-based Monitoring System

COVID-19 detection, fever detection, diagnosis of other infectious diseases, Indication of air pollution level in rooms

Yes

Yes

Yes

Yes

[Subection 5.4, Paragraph 5]

Table 1 presents the capabilities of our proposed system compared to other related systems. As it is clearly shown, our system can detect not only COVID-19 and other infectious diseases, but also air pollution level in different environments, e.g. rooms, halls, etc. Furthermore, our monitoring system provides security and privacy for its users, since it integrates many security features compared to other proposed schemes.

We provided a better description of our proposal by adding more text in the Sections that was short.

[Subsection 3.1]

3.1. Limitations could be addressed by IoT

Conventional medical methods have been effective in treating various health conditions, but they do have some limitations and barriers that can be addressed by integrating Internet of Things (IoT) and related technologies [9-11]. Some of these limitations and how IoT can help overcome them are [12] [13]:

Limited Monitoring: Conventional medical methods often involve periodic visits to healthcare facilities, which can lead to limited monitoring of patients' health conditions. IoT-enabled devices, such as wearable health trackers and remote patient monitoring systems, offer continuous and real-time health data collection, providing healthcare professionals with a more comprehensive view of patients' health statuses.

Inefficient Data Collection: The manual recording of patient information used in traditional data gathering techniques in the healthcare industry can be time-consuming and prone to inaccuracy. By automating data gathering procedures, IoT technologies lower the possibility of human error and facilitate quicker access to vital health data.

Lack of Personalization: Conventional medical practices frequently rely on generic methods that may not adequately address the needs of each patient. By evaluating a patient's real-time data, IoT can enable personalized medicine by allowing healthcare professionals to customize interventions and treatments depending on particular health problems and response patterns.

Limited Access to Healthcare in Remote Areas: There are many localities with poor access to healthcare facilities, particularly in rural or outlying areas. By enabling virtual consultations, remote diagnostics, and ongoing monitoring, IoT-enabled telemedicine and remote healthcare systems help close this gap, enhancing healthcare access and results in underserved areas.

Delayed Diagnosis and Intervention: Particularly in the case of chronic illnesses, traditional healthcare approaches may result in delayed diagnosis and action. IoT-based remote monitoring enables early abnormality detection, prompt response, and the avoidance of potential health emergencies.

Fragmented Healthcare Systems: Traditional medical practices frequently use disjointed data storage and communication platforms, which makes it difficult to access thorough patient records. Using IoT technologies, a more interconnected and unified healthcare system may be created by combining data from many sources and devices.

Patient Compliance and Adherence: It might be difficult to monitor and enforce patient adherence to recommended therapies and pharmaceutical regimens. IoT gadgets can deliver alerts, monitor medicine intake, and give behavioral cues, encouraging patients to follow their treatment regimens.

Limited Data for Research and Analysis: For medical research and analysis, traditional medical practices might only supply partial data. Big data analytics, medical research, and the creation of prognostic models for illness prevention and treatment are all greatly facilitated by IoT-generated data.

Data Security and Privacy Concerns: IoT use in healthcare sparks worries about patient privacy and data security. Strong data encryption, secure communication methods, and adherence to pertinent data protection laws like HIPAA and GDPR are all necessary for addressing these issues.

Healthcare may overcome these restrictions and obstacles by utilizing IoT and related technologies, which will result in better patient outcomes, improved disease management, and more effective healthcare delivery. When implementing IoT solutions in the healthcare industry, it is crucial to solve issues with data protection, interoperability, and scalability.

[Subsection 4.1]

4.1. Technological Challenges in Microbiology addressed by IoT

IoT (Internet of Things) and associated technologies can be very helpful in addressing a number of technological issues in the microbiology sector. The study of microorganisms, such as bacteria, viruses, fungus, and other tiny organisms, is known as microbiology [9] [12] [19]. IoT can assist in overcoming the following technological issues in micro-biology [11] [13]:

Real-Time Monitoring of Microbial Growth: Traditional techniques for keeping track of microbial development in cultures can be labor-intensive and time-consuming. Researchers can remotely monitor and control environmental conditions thanks to IoT-enabled sensors' ability to offer real-time data on variables like temperature, pH, and oxygen levels.

Remote Data Collection from Field Samples: It can be logistically difficult to gather field samples for microbiological investigation and transfer them to the lab. Researchers may track microbial activity on-site by using IoT-based remote sensing devices to collect data from varied environments and communicate it instantaneously to labs.

Data-Intensive Microbiome Analysis: Microbiome analysis, which examines microbial communities and their interactions throughout various ecosystems, produces a significant amount of data. High-throughput sequencing technology and IoT-based sensors can speed up data gathering and processing, which will improve our understanding of complex microbiomes.

Early Detection of Infectious Diseases: For the sake of public health, rapid and early detection of infectious diseases is essential. IoT devices can identify specific microbiological indicators, facilitating quicker diagnosis and intervention. Examples include wearable biosensors and point-of-care diagnostic tools.

Environmental Monitoring for Outbreaks: It might be difficult to monitor environmental conditions that cause illness epidemics, like air pollution and water quality. In order to identify potential outbreak hazards, IoT sensors may continuously monitor these elements and give real-time data.

Laboratory Automation and Efficiency: Microbiology laboratory processes can be labor- and time-intensive. Lab automation and IoT-connected hardware can automate sample processing, data collection, and analysis, increasing accuracy and efficiency.

Remote Collaboration and Data Sharing: The requirement for physical presence in the lab can make it difficult for researchers to work together on microbiology projects in various places. IoT technologies make it possible for researchers to securely access and exchange data online, facilitating remote collaboration.

Patient Monitoring and Management: In medical microbiology, it's important to keep an eye on individuals who have infections. IoT devices can help with ongoing patient monitoring by sending information regarding infection signs, the efficacy of treatments, and the progression of recovery to healthcare professionals.

Antibiotic Resistance Surveillance: Antibiotic resistance is being tracked and addressed, which is a challenge for world health. Real-time monitoring of antimicrobial use and resistance patterns by IoT-enabled equipment can help with surveillance and well-informed decision-making.

Quality Control in Food and Beverage Industry: Testing is required frequently to ensure microbiological safety in the manufacturing of food and beverages. IoT sensors can keep an eye on important parameters like temperature and cleanliness, preventing contamination and preserving product quality.

Data Security and Privacy: Strong security measures are needed while handling sensitive microbiological data to prevent unwanted access or data breaches. Encryption, secure communication protocols, and adherence to pertinent data protection laws should all be used in IoT solutions.

IoT and associated technologies, which allow real-time data collecting, automation, remote monitoring, and improved collaboration, present intriguing answers to these problems. However, data integrity, validation of IoT-generated data, and efficient integration with current laboratory operations should all be taken into account when implementing IoT in microbiology.

[Subsection 6.1]

6.1. IoT on Healthcare Applicaion Fields

IoT and other new technologies have a variety of uses in the healthcare industry that have the potential to completely change how care is delivered [9] [12]. The following are some important application areas where these technologies can have a big impact [11] [13]:

Remote Patient Monitoring: IoT devices allow for real-time data transmission to the cloud and continuous monitoring of patients' vital signs and health parameters. This is especially helpful for older patients, post-operative treatment, and controlling chronic illnesses.

Telemedicine and Virtual Health: IoT-driven telemedicine technologies make it easier to schedule follow-up appointments, remote diagnostics, and virtual consultations while maintaining data security.

Personalized Medicine: IoT devices gather patient-specific health data, which when combined with cloud-based analytics allows healthcare practitioners to customize interventions and treatments based on the needs of individual patients.

Disease Management and Prevention: IoT devices can help manage treatment regimens and track the evolution of diseases. Anomalies that are discovered early can assist avert complications and advance preventive healthcare.

Pharmaceutical Research and Manufacturing: IoT sensors improve quality control and regulatory compliance while ensuring ideal circumstances for medication development and manufacture.

Clinical Trials: The use of IoT devices increases the dependability of trial results and ensures the security of sensitive patient data by collecting precise, ongoing data from trial participants.

Elderly Care: Senior living facilities that use IoT-based monitoring systems encourage resident safety and wellbeing while securely storing data in the cloud for caregiver access.

Health and Wellness Wearables: IoT devices monitor physical activity, sleep habits, and general wellbeing, enabling people to actively control their health.

Emergency Response and Disaster Management: IoT-enabled medical equipment enables healthcare professionals to remotely assess patient conditions during emergencies and natural catastrophes, facilitating quick response.

Public Health Surveillance: IoT data gathering and analytics help to monitor population health trends, track disease outbreaks, and create efficient public health interventions.

Data-Driven Decision Making: When paired with cloud-based analytics and artificial intelligence, IoT-generated data helps healthcare managers and providers make decisions based on the best available evidence.

Healthcare Facility Management: IoT sensors track the health of medical equipment, the environment, and patient flow, enhancing both operational effectiveness and patient care.

Precision Agriculture and Nutrition: IoT devices keep an eye on crop health and soil conditions to help produce foods that are nutrient-rich and encourage preventive healthcare.

Medical Imaging and Diagnostics: Medical imaging equipment that is IoT-connected streamlines data collection and transfer, increasing the precision and speed of diagnostic procedures.

Medical Supply Chain Management: IoT sensors in the supply chain monitor the state of the equipment and medicines to ensure quality and avoid shortages.

Medical Training and Simulation: Medical education and skill development are improved by IoT-driven simulation systems that provide realistic training experiences.

Data Privacy and Security: Blockchain and other emerging technologies can improve the security and privacy of patient data, guaranteeing compliance with rules like HIPAA and GDPR.

IoT, emerging technologies, and secure cloud-based solutions have the power to revolutionize healthcare delivery, enhance patient outcomes, increase operational effectiveness, and spur innovation across a range of healthcare-related industries.

We additionally cited the following references you have suggested:

[6] V. K. Quy, A. Chehri, N. M. Quy, N. D. Han, N. T. Ban, “Innovative Trends in the 6G Era: A Comprehensive Survey of Architecture, Applications, Technologies, and Challenges”, IEEE Access, vol. 11, pp. 39824-39844, April 2023. [DOI: 10.1109/ACCESS.2023.3269297]

[7] V. K. Quy, N. V. Hau, D. V. Anh, L. A. Ngoc, “Smart healthcare IoT applications based on fog computing: archi-tecture, applications and challenges”, Springer, Complex & Intelligent Systems, vol. 8, pp. 3805-3815, November 2021. [DOI: 10.1007/s40747-021-00582-9]

[9] S. Shreya, K. Chatterjee, A. Singh, “A smart secure healthcare monitoring system with Internet of Medical Things”, Elsevier, Computer and Electrical Engineering, vo. 101, July 2022. [DOI: 10.1016/j.compeleceng.2022.107969]

[10] C. Butpheng, K.-H. Yeh, J.-L. Hou, “A Secure IoT and Cloud Computing-Enabled e-Health Management System”, Hindawi, Security and Communication Networks, vol. 2022, ID 300253, p. 14, June 2022. [DOI: 10.1155/2022/5300253]

[11] C. L. Stergiou, M. P. Koidou, K. E. Psannis, “IoT-Based Big Data Secure Transmission and Management over Cloud System: A Healthcare Digital Twin Scenario”, MDPI, Applied Sciences, vol. 13, issue: 16, pp. 9165, August 2023. [DOI: 10.3390/app13169165]

[12] L. Ogiela, A. Castiglione, B. B. Gupta, D. P. Agrawal, “IoT-based health monitoring system to handle pandemic diseases using estimated computing”, Springer, Neural Computing and Applications, vol. 35, pp. 13709-13710, May 2023. [DOI: 10.1007/s00521-023-08625-7]

[13] C. Butpheng, K.-H. Yeh, H. Xiong, “Security and Privacy in IoT-Cloud-Based e-Health Systems—A Comprehensive Review”, MDPI, Symmetry, vol. 12, issue: 7, pp. 1191, July 2020. [DOI: 10.3390/sym12071191]

[19] A. Singh, K. Chatterjee, “Edge computing based secure health monitoring framework for electronic healthcare system”, Springer, Cluster Computing, vol. 26, pp. 1205-1220, April 2023. [DOI: 10.1007/s10586-022-03717-w]

We have endeavored to rewrite the whole contributions part in order to be more clarify and clear.

We provided a better description of Subsection 5.1.1.. We have endeavored to extend the discussion of this part of our study in the following paragraphs.

[Subsection 5.1.1. Paragraphs 2 to 11]

Moreover, a Secure Monitoring System for IoT Healthcare Data on the Cloud is made possible thanks in large part to the Internet of Things (IoT). By connecting various gadgets and sensors to the internet, IoT technology enables data collection, transmission, and exchange. IoT devices can include wearable health monitors, smart medical equipment, remote patient monitoring systems, and more in the context of healthcare. Some major operations that directly relate the IoT to the healthcare sector are the following [4] [5] [8]:

  • Data Collection and Monitoring: IoT devices in the healthcare industry are able to continuously collect patient health information, including vital signs like heart rate, blood pressure, temperature, and glucose levels. These gadgets send the information to the cloud for centralized storage and supervision.
  • Real-time Data Streaming: Real-time data streaming is frequently provided by IoT devices, allowing healthcare providers and medical experts to view and track patients' health condition remotely. For the purpose of identifying anomalies and making any necessary interventions in a timely manner, real-time monitoring is essential.
  • Cloud-based Data Storage: To securely store the collected data, IoT healthcare devices often rely on cloud-based storage options. In order to ensure that patient health data is available from any location while protecting data integrity and confidentiality, cloud storage enables scalable and cost-effective data management.
  • Secure Data Transmission: It is critical to use secure communication protocols (like SSL/TLS) to encrypt data when it is being transported from IoT devices to the cloud. This guarantees data confidentiality and avoids illegal interception.
  • Authentication and Access Control: Strong authentication procedures are needed for a secure monitoring system for IoT healthcare data in order to confirm the legitimacy of users accessing the cloud system. Based on the user's role and rights, role-based access control (RBAC) can be used to limit data access.
  • Data Encryption and Decryption: Using encryption methods, such as the previously described Fernet encryption, guarantees that the private medical information is encrypted before being saved in the cloud. Only authorized users are permitted to decrypt files, protecting the secrecy of the data.
  • Monitoring and Anomaly Detection: To detect any strange patterns or potential security breaches in real-time, IoT devices and cloud systems can be fitted with monitoring and anomaly detection methods. As a result, the system is better able to react quickly to security problems.
  • Regulatory Compliance: Cloud-based IoT healthcare systems must abide by all applicable rules, including the GDPR, HIPAA, and other data protection legislation. Compliance guarantees patient confidentiality and safe handling of medical data.
  • Data Backup and Redundancy: In order to guarantee data accessibility and disaster recovery in the event of hardware breakdowns or system outages, cloud-based IoT systems can make use of the redundancy and backup options offered by the cloud provider.

Healthcare professionals can monitor patients remotely, continuously measure health metrics, and provide individualized care while ensuring data security and privacy by utilizing IoT technologies. The healthcare sector is enabled to take advantage of real-time data insights, improve patient outcomes, and improve overall healthcare service delivery by fusing IoT devices with a secure monitoring system in the cloud.

We provided a better description of Subsection 5.3 by adding more text where necessary. We have endeavored to extend the discussion of this part of our study in the following paragraphs.

[Subsection 5.3, Paragraphs 4, 5, 6, 7]

The system architecture for our suggested model implementation is depicted in Figure 2. Moreover, the operation of the proposed system of Figure 2 analyzed below. Each surveillance infrastructure is under the direction of the cloud server. An installed infrastructure that is not directly connected to a Cloud server may also provide data to it. To export the appropriate conclusions for each monitoring region, the Cloud server managed and organized all the data that the system had acquired. Additionally, the cloud offers a safe environment for gathering, managing, and processing the data generated by IoMT. Αll the peripheral equipment of the system is connected either to an autonomous network and through it to the Internet and from there to the proposed system, or is directly connected to the network of the proposed system.

Thermal-IR cameras measure the ambient temperature and, more precisely, the temperature of people arriving or moving through the surveillance area. Each camera sends pictures it has taken to a cloud server over a wireless network. A server in the cloud runs a data analysis algorithm that determines whether the temperature reading of each camera is greater than 37 Celsius. Then, in order to stop the spread and isolate that person in another, separated space, a notification signal appears in the monitoring system of that area if a temperature over that particular threshold is observed.

The area's environment has a high level of air pollution, according to air quality sensors. The cloud server received the detection data via the wireless network. The data analysis algorithm established on the Cloud server inspects if the rate of air pollution in the area exceeds normal. If it exceeds then a notification signal appears in the monitoring system of that area, and then the local surveillance and control personnel will be notified. The entire proposed system is based on the algorithm proposed below, and it involves all existing encryption/decryption models to create a new model that will provide direct data transmission and management in a secure Cloud environment.

A machine learning scenario is set up in the cloud server that will be able to detect and tell about the state of the monitored area. After many calculations and iterations, the ML algorithm will be able to export automated conclusions about whether or not a location is at risk. It manages and controls the data shared from the IoT sensors, both cameras and air sensors, as well as the data produced and collected by the connected mobile devices. It will therefore alert the local surveillance and control people based on the data given via the IoT sensors. The goal of the ML algorithm is to be able to detect without the need for human supervision and then properly notify about the detection it made.

We provided a better description of Algorithm 1 by adding more text. We have endeavored to extend the discussion of this part of our study in the following paragraph.

[Subsection 5.4, Paragraph 2]

This snippet of code shows how a Secure Monitoring System for IoT Healthcare Data may be implemented simply in the cloud. Therefore, the proposed code might not cover every element of a system used in production, thus will try to improve all the aspects of our proposed system through future work. Furthermore, we can clarify each element of it: Import Libraries: The code begins by importing necessary libraries, including ‘ssl’, ‘hashlib’, ‘Fernet’ from ‘cryptography.fernet’, and ‘socket’. IoT Device Data: A sample IoT device data dictionary (‘iot_data’) is created, simulating health-related information like patient ID, heart rate, and temperature. Encryption Key Generation: An encryption key is generated using Fernet from the ‘cryptography.fernet’ library. This key will be used for data encryption and decryption. Data Encryption: The ‘iot_data’ dictionary is converted to a string and encrypted using the generated encryption key, resulting in ‘encrypted_data’. Secure Communication Setup: An SSL/TLS secure communication context is created using the ‘ssl’ library, ensuring secure communication with the cloud server. Connect to Cloud Server: A secure connection is established with the cloud server using the given ‘cloud_server’ address and port number. Send Encrypted Data: The encrypted IoT data is sent to the cloud server using the secure connection. Receive and Validate Response: The code receives a response from the cloud server. The response is decrypted using the same encryption key. The code then compares the MD5 hash of the original response and the decrypted response to verify data integrity. Print Result: Based on the validation, the code prints whether the response integrity is verified or if tampering is detected. Close Connection: The secure connection to the cloud server is closed.

We have added two additional figures that depict our proposed system and its features. Please see Figures 1 & 4.

We have added Table 1 in which we compare the features of our proposed system with other relevant proposed systems.

The whole corrections or additions we have made in the paper we noticed with blue color.

Thank you very much for your attention and kind consideration.

Sincerely yours,

Round 2

Reviewer 1 Report

Comments and Suggestions for Authors

- All Figures should be enhanced with a resolution at a minimum of 300 dpi.

- Algorithm 1 should be rewritten to improve this work's quality.

Author Response

Dear reviewer,

We would like to thank you for your helpful comments on the paper. We appreciate the effort you put on, and the time you devoted to writing your suggestions. We believe, we have been able to address all your comments and correct any deficiencies you have pointed out.

  1. All Figures should be enhanced with a resolution at a minimum of 300 dpi.

Thank you for this advice. We appreciate your comment. All figures have been revised to a resolution at a minimum of 300 dpi.

  1. Algorithm 1 should be rewritten to improve this work's quality.

Thank you for your comments. We really appreciate it. We have revised the description of Algorithm 1 in a better way. Also we added more text in order to be described in a better way.

[Subsection 5.4, Paragraphs 2, 3, 4, 5]

This snippet of code shows how a Secure Monitoring System for IoT Healthcare Data may be implemented simply in the cloud. Therefore, the proposed code might not cover every element of a system used in production, thus will try to improve all the aspects of our proposed system through future work. Furthermore, we can clarify each element of it: Import Libraries: The code begins by importing necessary libraries, including ‘ssl’, ‘hashlib’, ‘Fernet’ from ‘cryptography.fernet’, and ‘socket’. IoT Device Data: A sample IoT device data dictionary (‘iot_data’) is created, simulating health-related information like patient ID, heart rate, and temperature. Encryption Key Generation: An encryption key is generated using Fernet from the ‘cryptography.fernet’ library. This key will be used for data encryption and decryption. Data Encryption: The ‘iot_data’ dictionary is converted to a string and encrypted using the generated encryption key, resulting in ‘encrypted_data’. Secure Communication Setup: An SSL/TLS secure communication context is created using the ‘ssl’ library, ensuring secure communication with the cloud server. Connect to Cloud Server: A secure connection is established with the cloud server using the given ‘cloud_server’ address and port number. Send Encrypted Data: The encrypted IoT data is sent to the cloud server using the secure connection. Receive and Validate Response: The code receives a response from the cloud server. The response is decrypted using the same encryption key. The code then compares the MD5 hash of the original response and the decrypted response to verify data integrity. Print Result: Based on the validation, the code prints whether the response integrity is verified or if tampering is detected. Close Connection: The secure connection to the cloud server is closed.

Moreover, a breakdown of the proposed algorithm briefly described here: 1) Regarding the imported libraries, ssl: Provides support for secure sockets using the SSL/TLS protocols. hashlib: Offers hash functions, which are used for integrity validation. socket: Enables communication over sockets. cryptography: A library for secure communication. cryptography.fernet.Fernet: A part of the cryptography library, used for symmetric (same key for encryption and decryption) encryption. 2) Regarding the IoT device data, a sample set of IoT device data (‘iot_data’) is created, representing health-related information like patient ID, heart rate, and temperature. 3) Regarding the Generate Encryption Key, an encryption key is generated using the Fernet symmetric encryption scheme from the “cryptography” library. 4) Regarding the Encrypt IoT Data, the IoT data is converted to a string, encoded, and then encrypted using the generated encryption key (‘cipher_suite’). The result is ‘encrypted_data’. 5) Regarding the Secure Communication Setup, an SSL/TLS secure communication context (context) is created. The purpose is set to “ssl.Purpose.related-client” (which might be a typo and could be “ssl.Purpose.SERVER_AUTH”), and certain verification checks are disabled to simplify the example. 6) Regarding the Connect to Cloud Server Securely, a secure socket is established (‘cloud_socket’) with the cloud server at a specified address (‘cloud_server’) and port (‘cloud_port’). The SSL/TLS wrap_socket method is used to secure the communication. 7) Regarding Send Encrypted Data to Cloud Server, the encrypted IoT data is sent to the cloud server over the secure connection. 8) Regarding Receive Response from Cloud Server, the script waits to receive a response from the cloud server. 9) Regarding Decrypt and Validate Response, the received response is decrypted using the same encryption key. Moreover, the integrity of the response is then validated by comparing the MD5 hash of the original response and the decrypted response. If the hashes match, the response integrity is considered verified; otherwise, a tampering detection message is printed. 10) Regarding Close the Connection, the secure connection to the cloud server is closed. Particularly, this algorithm is a basic illustration of secure communication practices, including data encryption and integrity verification, commonly used in IoT applications where maintaining the confidentiality and integrity of transmitted data is crucial.

The algorithm demonstrated in this Python source code scenario is a simplified example crafted for illustrative purposes, aiming to cover essential security considerations of a real-world implementation. For a comprehensive solution, additional measures such as robust error handling, advanced authentication mechanisms, secure storage practices, and compliance with relevant standards and regulations should be implemented.

This proposed source code algorithm illustrates the encryption of IoT data using the cryptography library and the utilization of SSL/TLS to establish a secure connection with the cloud server. Additionally, it incorporates a basic integrity check using the hashlib library to verify the response received from the server.

The whole corrections or additions we have made in the paper we noticed with blue color.

Thank you very much for your attention and kind consideration.

Sincerely yours,

Reviewer 3 Report

Comments and Suggestions for Authors

No further comments 

Author Response

Dear reviewer,

We would like to thank you for your helpful comments on the paper. We appreciate the effort you put on, and the time you devoted to writing your suggestions. We believe, we have been able to address all your comments and correct any deficiencies you have pointed out.

Moreover, bellow we mentioned further revisions that have been made before the last submission.

All figures have been revised to a resolution at a minimum of 300 dpi.

Thank you for your comments. We really appreciate it. We have revised the description of Algorithm 1 in a better way. Also we added more text in order to be described in a better way.

[Subsection 5.4, Paragraphs 2, 3, 4, 5]

This snippet of code shows how a Secure Monitoring System for IoT Healthcare Data may be implemented simply in the cloud. Therefore, the proposed code might not cover every element of a system used in production, thus will try to improve all the aspects of our proposed system through future work. Furthermore, we can clarify each element of it: Import Libraries: The code begins by importing necessary libraries, including ‘ssl’, ‘hashlib’, ‘Fernet’ from ‘cryptography.fernet’, and ‘socket’. IoT Device Data: A sample IoT device data dictionary (‘iot_data’) is created, simulating health-related information like patient ID, heart rate, and temperature. Encryption Key Generation: An encryption key is generated using Fernet from the ‘cryptography.fernet’ library. This key will be used for data encryption and decryption. Data Encryption: The ‘iot_data’ dictionary is converted to a string and encrypted using the generated encryption key, resulting in ‘encrypted_data’. Secure Communication Setup: An SSL/TLS secure communication context is created using the ‘ssl’ library, ensuring secure communication with the cloud server. Connect to Cloud Server: A secure connection is established with the cloud server using the given ‘cloud_server’ address and port number. Send Encrypted Data: The encrypted IoT data is sent to the cloud server using the secure connection. Receive and Validate Response: The code receives a response from the cloud server. The response is decrypted using the same encryption key. The code then compares the MD5 hash of the original response and the decrypted response to verify data integrity. Print Result: Based on the validation, the code prints whether the response integrity is verified or if tampering is detected. Close Connection: The secure connection to the cloud server is closed.

Moreover, a breakdown of the proposed algorithm briefly described here: 1) Regarding the imported libraries, ssl: Provides support for secure sockets using the SSL/TLS protocols. hashlib: Offers hash functions, which are used for integrity validation. socket: Enables communication over sockets. cryptography: A library for secure communication. cryptography.fernet.Fernet: A part of the cryptography library, used for symmetric (same key for encryption and decryption) encryption. 2) Regarding the IoT device data, a sample set of IoT device data (‘iot_data’) is created, representing health-related information like patient ID, heart rate, and temperature. 3) Regarding the Generate Encryption Key, an encryption key is generated using the Fernet symmetric encryption scheme from the “cryptography” library. 4) Regarding the Encrypt IoT Data, the IoT data is converted to a string, encoded, and then encrypted using the generated encryption key (‘cipher_suite’). The result is ‘encrypted_data’. 5) Regarding the Secure Communication Setup, an SSL/TLS secure communication context (context) is created. The purpose is set to “ssl.Purpose.related-client” (which might be a typo and could be “ssl.Purpose.SERVER_AUTH”), and certain verification checks are disabled to simplify the example. 6) Regarding the Connect to Cloud Server Securely, a secure socket is established (‘cloud_socket’) with the cloud server at a specified address (‘cloud_server’) and port (‘cloud_port’). The SSL/TLS wrap_socket method is used to secure the communication. 7) Regarding Send Encrypted Data to Cloud Server, the encrypted IoT data is sent to the cloud server over the secure connection. 8) Regarding Receive Response from Cloud Server, the script waits to receive a response from the cloud server. 9) Regarding Decrypt and Validate Response, the received response is decrypted using the same encryption key. Moreover, the integrity of the response is then validated by comparing the MD5 hash of the original response and the decrypted response. If the hashes match, the response integrity is considered verified; otherwise, a tampering detection message is printed. 10) Regarding Close the Connection, the secure connection to the cloud server is closed. Particularly, this algorithm is a basic illustration of secure communication practices, including data encryption and integrity verification, commonly used in IoT applications where maintaining the confidentiality and integrity of transmitted data is crucial.

The algorithm demonstrated in this Python source code scenario is a simplified example crafted for illustrative purposes, aiming to cover essential security considerations of a real-world implementation. For a comprehensive solution, additional measures such as robust error handling, advanced authentication mechanisms, secure storage practices, and compliance with relevant standards and regulations should be implemented.

This proposed source code algorithm illustrates the encryption of IoT data using the cryptography library and the utilization of SSL/TLS to establish a secure connection with the cloud server. Additionally, it incorporates a basic integrity check using the hashlib library to verify the response received from the server.

The whole corrections or additions we have made in the paper we noticed with blue color.

Thank you very much for your attention and kind consideration.

Sincerely yours,
